# Learning Curves for
# Deep Structured Gaussian Feature Models

**Jacob A. Zavatone-Veth**[1,2] and **Cengiz Pehlevan**[3,2,4]
[1]Department of Physics, [2]Center for Brain Science,
[3]John A. Paulson School of Engineering and Applied Sciences,
[4]Kempner Institute for the Study of Natural and Artificial Intelligence,
Harvard University
Cambridge, MA 02138, USA
jzavatoneveth@g.harvard.edu, cpehlevan@seas.harvard.edu

## Abstract

In recent years, significant attention in deep learning theory has been devoted to analyzing when models that interpolate their training data can still generalize well to unseen examples. Many insights have been gained from studying models with multiple layers of Gaussian random features, for which one can compute precise generalization asymptotics. However, few works have considered the effect of weight anisotropy; most assume that the random features are generated using independent and identically distributed Gaussian weights, and allow only for structure in the input data. Here, we use the replica trick from statistical physics to derive learning curves for models with many layers of structured Gaussian features. We show that allowing correlations between the rows of the first layer of features can aid generalization, while structure in later layers is generally detrimental. Our results shed light on how weight structure affects generalization in a simple class of solvable models.

## 1   Introduction

Characterizing how data structure and model architecture affect generalization performance is among the foremost goals of deep learning theory [1, 2]. A fruitful line of inquiry has focused on the properties of a class of simplified models that are asymptotically solvable: neural networks in which only the readout layer is trained and other weights are random, which are known as random feature models (RFMs) [3–21]. Though RFMs cannot capture the effects of representation learning on generalization in richly-trained neural networks [13, 22, 23], they have substantially advanced our understanding of how data structure and model architecture interact to give rise to a wide array of generalization phenomena observed in deep learning [1–5, 7–19, 24, 25].

Of particular interest is the question of when models overfit benignly, that is, when they generalize well despite having been trained to perfectly interpolate their training data. Here, much intuition has been gained by studying minimum-norm kernel interpolation—that is, the ridgeless limit of kernel ridge regression—with RFM kernels, for which precise generalization asymptotics can be computed using tools from random matrix theory. These asymptotics lead to a precise picture of how the spectrum of the random feature kernel and the structure of the task interact to determine generalization. These analyses are facilitated by universality results, often termed Gaussian equivalence theorems, that state that the generalization error of a nonlinear RFM is asymptotically equal to that of a linear Gaussian model with an effective noise term resulting from nonlinearity [3, 7, 10, 25, 26]. In the past few years, Gaussian equivalence theorems for ever more general classes of RFMs have been established: within this year Schröder et al. [20] and Bosch et al. [21] have established Gaussian equivalence theorems

37th Conference on Neural Information Processing Systems (NeurIPS 2023).

for deep nonlinear RFMs with unstructured feature weights, while Cui et al. [27] have extended some of these results to the setting of deep Bayesian neural networks when the target is of the same architecture.

However, these analyses consider the effect only of correlations in the data, and do not address the possibility of correlations between the random weights. It is standard to assume that the elements of the weight matrices at each layer are independent and identically distributed Gaussian random variables, and to our knowledge all existing Gaussian equivalence theorems make use of this assumption [3–15, 19–21]. As a result, how weight anisotropy affects generalization in deep RFMs—in particular, if it can affect the asymptotic scaling of generalization error with dataset size and network width [16, 19, 28]—remains unclear.

In this note, we take the first step towards filling that gap in our theoretical understanding of RFMs by computing the asymptotic generalization error of the simplest class of deep RFMs with anisotropic weight correlations: models with linear activations. Our primary contributions are as follows:

- Using the replica method from statistical mechanics [29], we compute the asymptotic generalization error of deep linear random feature models with weights drawn from general matrix Gaussian distributions. This computation is closely related to prior replica approches to product random matrix problems [13, 30].

- We show that, in the ridgeless limit, structure in the weights beyond the first layer is detrimental for generalization.

- We next consider the special case of power-law spectra in the weights and in the data, which was classically studied in kernel interpolation in the form of source-capacity conditions [31], and has recently attracted substantial interest in deep learning due to approximate power-law spectra present in real data [16, 19, 28, 32]. Using approximations for required spectral statistics derived in past works [19], we show that altering the power laws of the weight covariance spectra do not affect the scaling laws of generalization.

- We finally show how our results can be extended from the ridge regression estimator to the Bayesian Gibbs estimator, an object of classic study in the statistical physics of learning [13, 33, 34]. For sufficiently large prior variance, structure can be beneficial for generalization with this estimator.

Taken together, these results are consistent with the intuition that representation learning at only the first layer of a deep linear model is sufficient to recover a single teacher weight vector [13, 35–37].

## 2 Preliminaries

We consider depth-$L$ linear RFMs with input $\mathbf{x} \in \mathbb{R}^{n_0}$ and scalar output given by

$$g(\mathbf{x}; \mathbf{v}, \mathbf{F}) = \frac{1}{\sqrt{n_0}} (\mathbf{F}\mathbf{v})^\top \mathbf{x}, \tag{1}$$

where the feature matrix $\mathbf{F} \in \mathbb{R}^{n_0 \times n_L}$ is fixed and the vector $\mathbf{v} \in \mathbb{R}^{n_L}$ is trainable. If $L = 0$, corresponding to standard linear regression, the feature matrix is simply the identity: $\mathbf{F} = \mathbf{I}_{n_0}$. If $L > 0$, we take the feature matrix to be defined by a product of $L$ factors $\mathbf{U}_\ell \in \mathbb{R}^{n_{\ell-1} \times n_\ell}$:

$$\mathbf{F} = \frac{1}{\sqrt{n_1 \cdots n_L}} \mathbf{U}_1 \cdots \mathbf{U}_L. \tag{2}$$

We draw the random feature matrices independently from matrix Gaussian distributions

$$\mathbf{U}_\ell \sim \mathcal{MN}_{n_{\ell-1} \times n_\ell}(\mathbf{0}, \boldsymbol{\Gamma}_\ell, \boldsymbol{\Sigma}_\ell) \tag{3}$$

for input covariance matrices $\boldsymbol{\Gamma}_\ell \in \mathbb{R}^{n_{\ell-1} \times n_{\ell-1}}$ and output covariance matrices $\boldsymbol{\Sigma}_\ell \in \mathbb{R}^{n_\ell \times n_\ell}$, such that $\mathbb{E}[(U_\ell)_{ij}(U_{\ell'})_{i'j'}] = \delta_{\ell\ell'}(\Gamma_\ell)_{ii'}(\Sigma_\ell)_{jj'}$. Subject to the constraints of layer-wise independence and separability—which are required for the factors to be matrix-Gaussian distributed—this is the most general covariance structure one could consider. One might wish to relax this to include non-separable covariance tensors $\mathbb{E}[(U_\ell)_{ij}(U_{\ell'})_{i'j'}] = \delta_{\ell\ell'}(\chi_\ell)_{ii',jj'}$, but this would spoil the matrix-Gaussianity of the factors, and to our knowledge does not appear to be addressable using standard

methods [30, 38]. We generate training datasets according to a structured Gaussian covariate model, with $p$ i.i.d. training examples $(\mathbf{x}_\mu, y_\mu)$ generated as

$$\mathbf{x}_\mu \sim_{\text{i.i.d.}} \mathcal{N}(\mathbf{0}, \boldsymbol{\Sigma}_0), \qquad y_\mu = \frac{1}{\sqrt{n_0}} \mathbf{w}_*^\top \mathbf{x}_\mu + \xi_\mu, \tag{4}$$

where the teacher weight vector $\mathbf{w}_*$ is fixed and the label noise follows

$$\xi_\mu \sim_{\text{i.i.d.}} \mathcal{N}(0, \eta^2). \tag{5}$$

We collect the covariates into a matrix $\mathbf{X} \in \mathbb{R}^{p \times n_0}$, and the targets into a vector $\mathbf{y} \in \mathbb{R}^p$.

As in most works on RFMs [3–5, 8–21, 25], our focus is on the ridge regression estimator

$$\mathbf{v} = \arg\min_{\mathbf{v}} L \quad \text{for} \quad L = \frac{1}{2} \left\| \frac{1}{\sqrt{n_0}} \mathbf{X} \mathbf{F} \mathbf{v} - \mathbf{y} \right\|^2 + \frac{\lambda}{2} \|\boldsymbol{\Gamma}_{L+1}^{-1/2} \mathbf{v}\|_2^2, \tag{6}$$

where the positive-definite matrix $\boldsymbol{\Gamma}_{L+1} \in \mathbb{R}^{n_L \times n_L}$ controls the anisotropy of the norm and the ridge parameter $\lambda > 0$ sets the regularization strength. This minimization problem has the well-known closed form solution

$$\hat{\mathbf{v}} = \frac{1}{\sqrt{n_0}} \left( \lambda \boldsymbol{\Gamma}_{L+1}^{-1} + \frac{1}{n_0} \mathbf{F}^\top \mathbf{X}^\top \mathbf{X} \mathbf{F} \right)^{-1} \mathbf{F}^\top \mathbf{X}^\top \mathbf{y}. \tag{7}$$

As motivated in the Introduction, we are chiefly interested in the ridgeless limit $\lambda \downarrow 0$, in which the ridge regression solution gives the minimum $\ell_2$ norm interpolant of the training data. We measure performance of this estimator by the generalization error

$$\epsilon_{p,n_0,\ldots,n_L} = \mathbb{E}_{\mathbf{x}} \left( g(\mathbf{x}; \hat{\mathbf{v}}, \mathbf{F}) - \mathbb{E}_\xi[y(\mathbf{x})] \right)^2 = \frac{1}{n_0} \|\boldsymbol{\Sigma}_0^{1/2} (\mathbf{F} \hat{\mathbf{v}} - \mathbf{w}_*)\|^2, \tag{8}$$

which is a random variable with distribution induced by the training data and feature weights.

This leads us to a simple, but important observation: including structured input-input covariances is equivalent to transforming the feature-feature covariances. We state this formally as:

**Lemma 2.1.** *Fix sets of matrices $\{\boldsymbol{\Gamma}_\ell\}_{\ell=1}^{L+1}$ and $\{\boldsymbol{\Sigma}_\ell\}_{\ell=0}^{L}$, and a target vector $\mathbf{w}_*$. Let $\epsilon_{p,n_0,\ldots,n_L}$ be the resulting generalization error as defined in* (8). *Let*

$$\tilde{\boldsymbol{\Gamma}}_\ell = \mathbf{I}_{n_{\ell-1}} \qquad\qquad\qquad \text{for } \ell = 1, \ldots, L+1, \tag{9}$$

$$\tilde{\boldsymbol{\Sigma}}_\ell = \boldsymbol{\Gamma}_{\ell+1}^{1/2} \boldsymbol{\Sigma}_\ell \boldsymbol{\Gamma}_{\ell+1}^{1/2} \qquad\qquad \text{for } \ell = 0, \ldots, L, \text{ and} \tag{10}$$

$$\tilde{\mathbf{w}}_* = \boldsymbol{\Gamma}_1^{-1/2} \mathbf{w}_*. \tag{11}$$

*Let $\tilde{\epsilon}_{p,n_0,\ldots,n_L}$ be the generalization error for these transformed covariance matrices and target. Then, for any $\lambda > 0$, we have the equality in distribution $\epsilon_{p,n_0,\ldots,n_L} \stackrel{d}{=} \tilde{\epsilon}_{p,n_0,\ldots,n_L}$.*

*Proof of Lemma 2.1.* As the features and data are Gaussian, we can write $\mathbf{X} \stackrel{d}{=} \boldsymbol{\Sigma}_0^{1/2} \mathbf{Z}_0$ and $\mathbf{U}_\ell \stackrel{d}{=} \boldsymbol{\Gamma}_\ell^{1/2} \mathbf{Z}_\ell \boldsymbol{\Sigma}_\ell^{1/2}$ for unstructured Gaussian matrices $(Z_\ell)_{ij} \sim_{\text{i.i.d.}} \mathcal{N}(0,1)$. Substituting these representations into the ridge regression solution (7) and the generalization error (8), the claim follows. $\square$

Therefore, we may take $\boldsymbol{\Gamma}_\ell = \mathbf{I}_{n_{\ell-1}}$ without loss of generality. Moreover, thanks to the rotation-invariance of the isotropic Gaussian factors $\mathbf{Z}_\ell$, we may in fact take the remaining covariance matrices $\boldsymbol{\Sigma}_\ell$ to be diagonal without loss of generality, so long as we then express $\tilde{\mathbf{w}}_*$ in the basis of eigenvectors of $\boldsymbol{\Sigma}_0$. An important qualitative takeaway of this result is that changing the covariance matrix of the inputs of the first layer $\boldsymbol{\Gamma}_1$ is equivalent to modifying the data covariance matrix, which was in a simpler form observed in the shallow setting ($L = 1$) by Pandey et al. [39].

## 3 Asymptotic learning curves

Having defined the setting of our problem, we can define our concrete objective and state our main results, deferring their interpretation to the following section. We consider the standard proportional asymptotic limit

$$p, n_0, \ldots, n_L \to \infty, \quad \text{with} \quad n_\ell/p \to \alpha_\ell \in (0, \infty), \tag{12}$$

which we will refer to as the thermodynamic limit. Our goal is to compute the limiting generalization error:

$$\epsilon = \lim_{p,n_0,\dots,n_L \to \infty} \mathbb{E}_{\mathcal{D}} \frac{1}{n_0} \|\mathbf{\Sigma}_0^{1/2}(\mathbf{F}\mathbf{v} - \mathbf{w}_*)\|^2, \tag{13}$$

where $\mathbb{E}_{\mathcal{D}}$ denotes expectation over all sources of quenched disorder in the problem, i.e., the training data and the random feature weights. In the thermodynamic limit, we expect the generalization error to concentrate, which is why we compute its average in (13) [3–5, 8–21].

To have a well-defined thermodynamic limit, the covariances $\tilde{\mathbf{\Sigma}}_\ell$ and the teacher $\tilde{\mathbf{w}}_\ell$ must be in some sense sufficiently well-behaved. We consider the following conditions, which are the generalization to our setting of those assumed in previous work [4–7, 16–18, 40]:

**Assumption 3.1.** *We assume that we are given deterministic sequences of positive-definite matrices $\tilde{\mathbf{\Sigma}}_\ell(n_\ell)$ and vectors $\tilde{\mathbf{w}}_*(n_0)$ indexed by the system size, such that the limiting (weighted) spectral moment generating functions*

$$M_{\tilde{\mathbf{\Sigma}}_\ell}(z) = \lim_{n_\ell \to \infty} \frac{1}{n_\ell} \operatorname{tr}[\tilde{\mathbf{\Sigma}}_\ell(z\mathbf{I}_{n_\ell} - \tilde{\mathbf{\Sigma}}_\ell)^{-1}] \quad and \quad \psi(z) = \lim_{n_0 \to \infty} \frac{1}{n_0} \tilde{\mathbf{w}}_*^\top \tilde{\mathbf{\Sigma}}_0(z\mathbf{I}_{n_0} + \tilde{\mathbf{\Sigma}}_0)^{-1} \tilde{\mathbf{w}}_* \tag{14}$$

*are well-defined, for all $\ell = 0, \dots, L$.*

We can now state our results. As a preliminary step, we first give an expression for the generalization error for a fixed teacher $\tilde{\mathbf{w}}_*$ at finite ridge $\lambda$. Then, we pass to the ridgeless limit, on which we focus for the remainder of the paper. At finite ridge, we have the following:

**Proposition 3.1.** *Assume Assumption 3.1 holds. For $\lambda > 0$, let $\zeta$ solve the self-consistent equation*

$$\lambda = \frac{1-\zeta}{\zeta} \prod_{\ell=0}^{L} \frac{-\zeta}{\alpha_\ell} M_{\tilde{\mathbf{\Sigma}}_\ell}^{-1} \left(-\frac{\zeta}{\alpha_\ell}\right). \tag{15}$$

*In terms of $\zeta$, let $\kappa_\ell(\zeta)$ solve*

$$\mathbb{E}_{\tilde{\sigma}_\ell} \left[\frac{\tilde{\sigma}_\ell}{\kappa_\ell(\zeta) + \tilde{\sigma}_\ell}\right] = -M_{\tilde{\mathbf{\Sigma}}_\ell}(-\kappa_\ell(\zeta)) = \frac{\zeta}{\alpha_\ell} \tag{16}$$

*for $\ell = 0, \dots, L$, where $\mathbb{E}_{\tilde{\sigma}_\ell}[h(\tilde{\sigma}_\ell)] = \lim_{n_\ell \to \infty} n_\ell^{-1} \sum_{j=1}^{n_\ell} h(\tilde{\sigma}_{\ell,j})$ denotes expectation of a function $h$ with respect to the limiting spectral distribution of $\tilde{\mathbf{\Sigma}}_\ell$, for $\tilde{\sigma}_{\ell,j}$ its eigenvalues at finite size, and let*

$$\mu_\ell(\zeta) = -\frac{\alpha_\ell}{\zeta} \kappa_\ell(\zeta) M'_{\tilde{\mathbf{\Sigma}}_\ell}(-\kappa_\ell(\zeta)) = 1 - \frac{\alpha_\ell}{\zeta} \mathbb{E}_{\tilde{\sigma}_\ell} \left[\left(\frac{\tilde{\sigma}_\ell}{\kappa_\ell(\zeta) + \tilde{\sigma}_\ell}\right)^2\right]. \tag{17}$$

*Then, the learning curve (13) at finite ridge for a fixed target is given by*

$$\left[1 + \left(\sum_{\ell=0}^{L} \frac{1-\mu_\ell}{\mu_\ell}\right)(1-\zeta)\right]\epsilon = \left(\sum_{\ell=1}^{L} \frac{1-\mu_\ell}{\mu_\ell}\right)\kappa_0 \psi(\kappa_0) - \frac{\kappa_0^2}{\mu_0}\psi'(\kappa_0) + \left(\sum_{\ell=0}^{L} \frac{1-\mu_\ell}{\mu_\ell}\right)\zeta\eta^2. \tag{18}$$

*Proof of Proposition 3.1.* We defer the derivation of (18) to Appendix A. To compute the disorder average in (13), we express the minimization problem in (6) as the zero-temperature limit $\beta \to \infty$ of an auxiliary Gibbs distribution $p(\mathbf{v}) \propto e^{-\beta L}$, and evaluate the average over the random data random feature weights using the non-rigorous replica method from the statistical mechanics of disordered systems [29, 33]. This computation is lengthy but standard, and is closely related to the approach used in our previous works on deep linear models [13, 30]. All of our results are obtained under a replica-symmetric *Ansatz*; as the ridge regression problem (6) is convex, we expect replica symmetry to be unbroken [29, 41, 42]. $\square$

From the self-consistent equation (15), we recognize that $\zeta$ is is up to a sign the spectral moment generating function of the feature Gram matrix $\mathbf{K} = \mathbf{X}\mathbf{F}\mathbf{F}^\top\mathbf{X}^\top/n_0$, which is a product-Wishart random matrix [30]:

$$\zeta(\lambda) = -M_{\mathbf{K}}(-\lambda). \tag{19}$$

This dependence falls out of the replica computation of the generalization error using an auxiliary Gibbs distribution; we emphasize that one could take an alternative approach in which the generalization error is first expressed in terms of $M_{\mathbf{K}}$—as, for instance, in Gerace et al. [25] or Hastie et al.

[5]—and then use results on the spectra of product-Wishart matrices to conclude the claimed result [30]. This approach would potentially have the advantage of giving a fully rigorous proof, rather than one that depends on the replica trick. However, one would still then be faced with the task of solving the self-consistent equation for the spectral moment generating function, and therefore would end up in the same place insofar as quantitative predictions are concerned.

In principle, we could now directly proceed to study how weight structure affects (18) for some fixed ridge $\lambda$. However, as long as there is structure in the weights and/or the data, the self-consistent equation (15) must generally be solved numerically [14, 30]. To allow us to make analytical progress, we therefore focus on the ridgeless limit $\lambda \downarrow 0$ for the remainder of the present paper, and leave careful analysis of the $\lambda > 0$ case to future work. This follows the path of most recent studies of models with linear random features, and also the fundamental interest in interpolating models [3–17, 19–21]. We therefore emphasize that we state Proposition 3.1 merely as a preliminary result.

Before giving our result for the generalization error in the ridgeless limit, we warn the reader of an impending, somewhat severe abuse of notation: in Proposition 3.2 and for the remainder of the paper, we will re-define $\kappa_\ell$ to be given by its value for the solution for $\zeta$ appropriate in the regime of interest. Moreover, we will simply write $\epsilon$ for $\lim_{\lambda \downarrow 0} \epsilon$.

**Proposition 3.2.** *Assume Assumption 3.1 holds, and let $\alpha_{\min} = \min\{\alpha_1, \cdots, \alpha_L\}$. For $\ell = 0, \ldots, L$, in the regime $\alpha_\ell > 1$, let $\kappa_\ell$ be given by the unique non-negative solution to the implicit equation*

$$\frac{1}{\alpha_\ell} = -M_{\tilde{\boldsymbol{\Sigma}}_\ell}(-\kappa_\ell) = \mathbb{E}_{\tilde{\sigma}_\ell}\left[\frac{\tilde{\sigma}_\ell}{\kappa_\ell + \tilde{\sigma}_\ell}\right]. \tag{20}$$

*In terms of $\kappa_\ell$, let*

$$\mu_\ell = -\alpha_\ell \kappa_\ell M'_{\tilde{\boldsymbol{\Sigma}}_\ell}(-\kappa_\ell) = 1 - \alpha_\ell \mathbb{E}_{\tilde{\sigma}_\ell}\left[\left(\frac{\tilde{\sigma}_\ell}{\kappa_\ell + \tilde{\sigma}_\ell}\right)^2\right]. \tag{21}$$

*In the regime $\alpha_{\min} < \alpha_0$, let $\kappa_{\min}$ be the unique non-negative solution to the implicit equation*

$$\frac{\alpha_{\min}}{\alpha_0} = -M_{\tilde{\boldsymbol{\Sigma}}_0}(-\kappa_{\min}) = \mathbb{E}_{\tilde{\sigma}_0}\left[\frac{\tilde{\sigma}_0}{\kappa_{\min} + \tilde{\sigma}_0}\right]. \tag{22}$$

*Then, the learning curve (13) for a fixed target in the ridgeless limit $\lambda \downarrow 0$ is given by*

$$\epsilon = \begin{cases} \left(\sum_{\ell=1}^L \frac{1-\mu_\ell}{\mu_\ell}\right)\kappa_0 \psi(\kappa_0) - \frac{\kappa_0^2}{\mu_0}\psi'(\kappa_0) + \left(\sum_{\ell=0}^L \frac{1-\mu_\ell}{\mu_\ell}\right)\eta^2, & \alpha_0, \alpha_{\min} > 1 \\ \frac{\kappa_{\min}\psi(\kappa_{\min})}{1-\alpha_{\min}} + \frac{\alpha_{\min}}{1-\alpha_{\min}}\eta^2, & \alpha_{\min} < 1, \alpha_{\min} < \alpha_0 \\ \frac{\alpha_0}{1-\alpha_0}\eta^2, & \alpha_0 < 1, \alpha_0 < \alpha_{\min}. \end{cases} \tag{23}$$

*Proof of Proposition 3.2.* We derive (23) as the zero-ridge limit of Proposition 3.1 in Appendix A. $\qquad\square$

Before we analyze the effect of weight anisotropy in detail in Section 4, we note several simplifying special cases of Proposition 3.2 which recover the results of prior works. To facilitate this comparison, we provide a notational dictionary in Appendix D. The first important special case is

**Corollary 3.1.** *If $L = 0$, we have*

$$\epsilon = \begin{cases} -\frac{\kappa_0^2}{\mu_0}\psi'(\kappa_0) + \frac{1-\mu_0}{\mu_0}\eta^2, & \alpha_0 > 1 \\ \frac{\alpha_0}{1-\alpha_0}\eta^2, & \alpha_0 < 1. \end{cases} \tag{24}$$

This recovers the known, rigorously proved result for linear ridgeless regression [4–7, 16–18]. For larger depths, an important simplifying case of Proposition 3.2 is that in which the data and features are unstructured, in which case the generalization error is given by

**Corollary 3.2.** *If $\tilde{\boldsymbol{\Sigma}}_\ell = \mathbf{I}_{n_\ell}$ for $\ell = 0, \ldots, L$, we have, for any target satisfying $\|\tilde{\mathbf{w}}_*\|^2 = n_0$,*

$$\epsilon = \begin{cases} \left(1 + \sum_{\ell=1}^L \frac{1}{\alpha_\ell - 1}\right)\left(1 - \frac{1}{\alpha_0}\right) + \left(\sum_{\ell=0}^L \frac{1}{\alpha_\ell - 1}\right)\eta^2, & \alpha_0, \alpha_{\min} > 1 \\ \frac{1-\alpha_{\min}/\alpha_0}{1-\alpha_{\min}} + \frac{\alpha_{\min}}{1-\alpha_{\min}}\eta^2, & \alpha_{\min} < 1, \alpha_{\min} < \alpha_0 \\ \frac{\alpha_0}{1-\alpha_0}\eta^2, & \alpha_0 < 1, \alpha_0 < \alpha_{\min}. \end{cases} \tag{25}$$

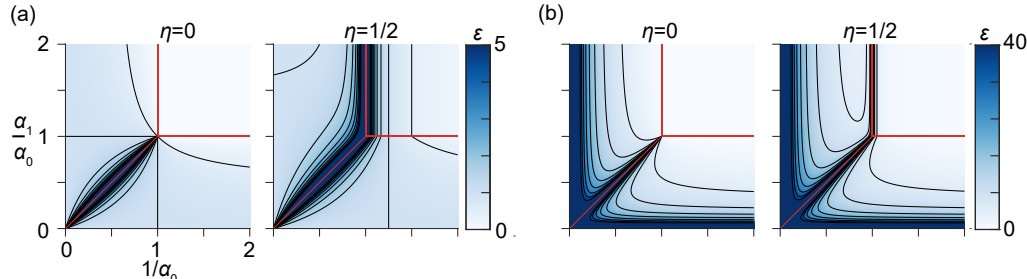

Figure 1: Phase diagram of generalization in deep linear RFMs. For simplicity, we consider a model with a single hidden layer ($L = 1$); the picture for deeper models is identical if one considers the narrowest hidden layer [13]. (a). Generalization error $\epsilon$ for unstructured data and features from (25) as a function of training data density $1/\alpha_0$ and hidden layer width $\alpha_1/\alpha_0$ in the absence of label noise ($\eta = 0$; *left*) and in the presence of label noise ($\eta = 0.5$; *right*). (b). As in (a), but for power law structured data and weights, with $\omega_0 = \omega_1 = 1$, and $\bar{\epsilon}$ given by (31). See Appendix F for numerical methods.

*Proof of Corollary 3.2.* We have $M_{\mathbf{I}_{n_\ell}}(z) = 1/(z-1)$, hence $\kappa_\ell = \alpha_\ell - 1$, $\mu_\ell = 1 - 1/\alpha_\ell$, and $\kappa_{\min} = \alpha_0/\alpha_{\min} - 1$. Finally, for any fixed teacher vector satisfying $\|\tilde{\mathbf{w}}_*\|^2 = n_0$, we have $\psi(z) = 1/(z+1)$ if $\tilde{\mathbf{\Sigma}}_0 = \mathbf{I}_{n_0}$. Substituting these results into (23), we obtain (25). $\square$

This recovers the result obtained in our previous work [13], and in the single-layer case $L = 1$ recovers results obtained by Rocks and Mehta [14, 15], and by Hastie et al. [5] (see Appendix D). In the slightly more general case of unstructured weights but structured features, we have

**Corollary 3.3.** *If $\tilde{\mathbf{\Sigma}}_\ell = \mathbf{I}_{n_\ell}$ for $\ell = 1, \ldots, L$, but $\tilde{\mathbf{\Sigma}}_0 \neq \mathbf{I}_{n_0}$, we have, for any target satisfying $\|\tilde{\mathbf{w}}_*\|^2 = n_0$,*

$$
\epsilon = \begin{cases} \left(\sum_{\ell=1}^L \frac{1}{\alpha_\ell-1}\right)\kappa_0\psi(\kappa_0) - \frac{\kappa_0^2}{\mu_0}\psi'(\kappa_0) + \left(\frac{1-\mu_0}{\mu_0} + \sum_{\ell=1}^L \frac{1}{\alpha_\ell-1}\right)\eta^2, & \alpha_0, \alpha_{\min} > 1 \\ \frac{\kappa_{\min}\psi(\kappa_{\min})}{1-\alpha_{\min}} + \frac{\alpha_{\min}}{1-\alpha_{\min}}\eta^2, & \alpha_{\min} < 1, \alpha_{\min} < \alpha_0 \\ \frac{\alpha_0}{1-\alpha_0}\eta^2, & \alpha_0 < 1, \alpha_0 < \alpha_{\min}. \end{cases}
\tag{26}
$$

*Proof of Corollary 3.3.* (26) follows from substituting the results of Corollary 3.2 into (23). $\square$

In the special case $L = 1$, this recovers the result obtained using rigorous methods in contemporaneous work by Bach [40], posted to the arXiv one day after the first version of our work [43]. Here, as the data spectrum and target vector enter the generalization error in nearly the same way as in the case of linear regression, all of the intuitions developed in that case can be carried over [4–7, 16–18].

Another useful simplification can be obtained by further averaging over isotropically-distributed teachers $\tilde{\mathbf{w}}_* \sim \mathcal{N}(\mathbf{0}, \mathbf{I}_{n_0})$, which gives

**Corollary 3.4.** *Let $\bar{\epsilon} = \mathbb{E}_{\tilde{\mathbf{w}}_* \sim \mathcal{N}(\mathbf{0}, \mathbf{I}_{n_0})}[\epsilon]$. Then, we have*

$$
\bar{\epsilon} = \begin{cases} \left(1 + \sum_{\ell=1}^L \frac{1-\mu_\ell}{\mu_\ell}\right)\frac{\kappa_0}{\alpha_0} + \left(\sum_{\ell=0}^L \frac{1-\mu_\ell}{\mu_\ell}\right)\eta^2, & \alpha_0, \alpha_{\min} > 1 \\ \frac{\alpha_{\min}\kappa_{\min}/\alpha_0}{1-\alpha_{\min}} + \frac{\alpha_{\min}}{1-\alpha_{\min}}\eta^2, & \alpha_{\min} < 1, \alpha_{\min} < \alpha_0 \\ \frac{\alpha_0}{1-\alpha_0}\eta^2, & \alpha_0 < 1, \alpha_0 < \alpha_{\min}. \end{cases}
\tag{27}
$$

*Proof of Corollary 3.4.* Observing that $\mathbb{E}_{\tilde{\mathbf{w}}_*}\psi(z) = -M_{\tilde{\mathbf{\Sigma}}_0}(-z)$, the claim follows from (23). $\square$

In the special case of a single layer of unstructured feature weights ($L = 1$, $\tilde{\mathbf{\Sigma}}_1 = \mathbf{I}_{n_1}$), this recovers the result of recent work by Maloney et al. [19], who used a planar diagram method to the generalization error of single-hidden-layer linear RFMs with unstructured weights (see Appendix D).

Another important simplifying case of Proposition 3.2 is the limit in which the hidden layer widths are large, in which the generalization error of the deep RFM reduces to that of a shallow model, as given by Corollary 3.1. More precisely, we have a large-width expansion given by:

**Corollary 3.5.** *In the large-width regime $\alpha_1, \ldots, \alpha_L \gg 1$, assuming that the weight spectra have finite moments, the generalization error* (23) *expands as*

$$\epsilon = -\frac{\kappa_0^2}{\mu_0}\psi'(\kappa_0) + \frac{1-\mu_0}{\mu_0}\eta^2 + \big(\sum_{\ell=1}^{L} \frac{\mathbb{E}_{\tilde{\sigma}_\ell}[\tilde{\sigma}_\ell^2]}{\mathbb{E}_{\tilde{\sigma}_\ell}[\tilde{\sigma}_\ell]^2} \frac{1}{\alpha_\ell}\big)(\kappa_0\psi(\kappa_0) + \eta^2) + \mathcal{O}(\alpha_1^{-2}, \ldots, \alpha_L^{-2}) \quad (28)$$

*in the regime $\alpha_0 > 1$; if $\alpha_0 < 1$ the generalization error does not depend on the hidden layer widths so long as they are greater than 1.*

*Proof of Corollary 3.5.* See Appendix E. $\qquad\square$

# 4 How does weight structure affect generalization?

The first salient feature of the learning curves given by Proposition 3.2 is that the addition of weight structure does not alter the phase diagram of generalization, which is illustrated in Figure 1. There are three qualitatively distinct phases present, depending on the data density and minimum layer width: the overparameterized regime $\alpha_0, \alpha_{\min} > 1$, the bottlenecked regime $\alpha_{\min} < 1$, $\alpha_{\min} < \alpha_0$, and the overdetermined regime $\alpha_0 < 1$, $\alpha_0 < \alpha_{\min}$. This dependence on the narrowest hidden layer matches our previous work on models with unstructured weights [13][1], and can be observed in the solutions to the ridge regression problem for fixed data (Appendix C). As $\alpha_\ell \downarrow 1$, $\kappa_\ell \downarrow 0$ and $\mu_\ell \downarrow 0$, and the generalization error diverges. Similarly, the generalization error diverges as $\alpha_{\min} \uparrow 1$, or $\alpha_0 \uparrow 1$ in the presence of label noise. However, there are not multiple descents in these deep linear models, consistent with the qualitative picture of the effect of nonlinearity given by previous works [9, 10].

The second salient feature of Proposition 3.2 is that the matrices $\tilde{\Sigma}_\ell$ enter the generalization error independently; there are no 'interaction' terms involving products of the correlation matrices for different layers. This decoupling is expected given that the features are Gaussian and independent across layers [30]. Moreover, under the rescaling $\tilde{\Sigma}'_\ell = \tau_\ell \tilde{\Sigma}_\ell$ for $\tau_\ell > 0$, we have $\kappa'_\ell = \tau_\ell \kappa_\ell$ and $\mu'_\ell = \mu_\ell$ (we show this explicitly in Appendix B). Therefore, (23) is sensitive only to the overall scale of $\tilde{\Sigma}_0$, not to the scales of $\tilde{\Sigma}_1, \ldots, \tilde{\Sigma}_L$. This scale-invariance can be observed directly from the ridgeless limit of the ridge regression estimator (7).

We can gain intuition for the effect of having $\tilde{\Sigma}_\ell \not\propto \mathbf{I}_{n_\ell}$ for $\ell \geq 1$ through the following argument:

**Lemma 4.1.** *Under the conditions of Proposition 3.2, in the regime $\alpha_0, \alpha_{\min} > 1$, we have*

$$\epsilon \geq \big(\sum_{\ell=1}^{L} \frac{1}{\alpha_\ell - 1}\big)\kappa_0\psi(\kappa_0) - \frac{\kappa_0^2}{\mu_0}\psi'(\kappa_0) + \big(\frac{1-\mu_0}{\mu_0} + \sum_{\ell=1}^{L} \frac{1}{\alpha_\ell - 1}\big)\eta^2. \quad (29)$$

*That is, the generalization error for a given $\tilde{\Sigma}_1, \cdots, \tilde{\Sigma}_L$ is bounded from below by the generalization error for $\tilde{\Sigma}_\ell = \mathbf{I}_{n_\ell}$ for $\ell = 1, \ldots, L$.*

*Proof of Lemma 4.1.* In Appendix B, we show that $\mu_\ell \leq 1 - 1/\alpha_\ell$ for any weight spectrum, which implies that $(1 - \mu_\ell)/\mu_\ell \geq 1/(\alpha_\ell - 1)$. Substituting these bounds in to the general expression for the generalization error in this regime from (23), the claim follows. $\qquad\square$

Therefore, having $\tilde{\Sigma}_\ell \neq \mathbf{I}_{n_\ell}$ for $\ell = 1, \ldots, L$ cannot improve generalization in the $\alpha_0, \alpha_{\min} > 1$ regime. This is consistent with the large-width expansion in Corollary 3.5, where we can apply Jensen's inequality to bound the weight-dependence of the correction as $\mathbb{E}_{\tilde{\sigma}_\ell}[\tilde{\sigma}_\ell^2]/\mathbb{E}_{\tilde{\sigma}_\ell}[\tilde{\sigma}_\ell]^2 \geq 1$, with equality only when the weights are unstructured. In other regimes, $\tilde{\Sigma}_1, \cdots, \tilde{\Sigma}_L$ do not affect the generalization error. In contrast, a similar argument shows that anisotropy in $\tilde{\Sigma}_0$ can be beneficial in the target-averaged case, at least in the absence of label noise. We formalize this as:

**Lemma 4.2.** *Under the conditions of Corollary 3.4, in the absence of label noise ($\eta = 0$), we have*

$$\bar{\epsilon} \leq \begin{cases} \big(1 + \sum_{\ell=1}^{L} \frac{1-\mu_\ell}{\mu_\ell}\big)\big(1 - \frac{1}{\alpha_0}\big)\mathbb{E}[\tilde{\sigma}_0], & \alpha_0, \alpha_{\min} > 1 \\ \frac{(1-\alpha_{\min}/\alpha_0)}{1-\alpha_{\min}}\mathbb{E}[\tilde{\sigma}_0], & \alpha_{\min} < 1, \alpha_{\min} < \alpha_0 \\ 0, & \alpha_0 < 1, \alpha_0 < \alpha_{\min}. \end{cases} \quad (30)$$

*That is, $\bar{\epsilon}$ for a given $\tilde{\Sigma}_0$ is bounded from above by the generalization error for a flat spectrum $\tilde{\Sigma}_0 = \mathbb{E}[\tilde{\sigma}_0]\mathbf{I}_{n_0}$.*

---

[1]Previous works on deep RFMs have used several different parameterizations of the thermodynamic limit [3–17, 19–21]. We detail the conversion between these conventions in Appendix D.

*Proof of Lemma 4.2.* In Appendix B, we show that $\kappa_0 \leq (\alpha_0 - 1)\mathbb{E}[\tilde{\sigma}_0]$. As its defining equation (22) is of the same form as (20), the corresponding bound for $\kappa_{\min}$ follows immediately: $\kappa_{\min} \leq (\alpha_0/\alpha_{\min} - 1)\mathbb{E}[\tilde{\sigma}_0]$. Substituting these bounds into (27) with $\eta = 0$, the claim follows. $\square$

If $\mathbb{E}[\tilde{\sigma}_0]$ is not finite, then this bound is entirely vacuous: $\bar{\epsilon} \leq \infty$. If we do not average over isotropically-distributed targets, then the effect of anisotropy in $\tilde{\Sigma}_0$ is harder to analyze. Previous works have, however, analyzed the interaction of data structure with a fixed target in great detail for models with $L = 0$ or $L = 1$, showing that targets that align with the top eigenvectors of $\tilde{\Sigma}_0$ are easier to learn [5, 16, 17, 42, 44].

## 5 Power law spectra

We can gain further intuition for the effect of weight structure by considering an approximately solvable model for anisotropic spectra: power laws [16, 19, 28]. Power law data spectra have recently attracted considerable attention as a possible model for explaining the scaling laws of generalization observed in large language models [16, 19, 28, 32]. Maloney et al. [19] proposed a single-hidden-layer ($L = 1$) linear RFM with power-law-structured data and unstructured weights as a model for neural scaling laws. Does introducing power law structure into the weights affect the scaling laws predicted by deep linear RFMs? We have the following result:

**Corollary 5.1.** *At finite size, define each covariance matrix $\tilde{\Sigma}_\ell$ such that its $j$-th eigenvalue is $\tilde{\sigma}_{\ell,j} = \tilde{\varsigma}_\ell(n_\ell/j)^{1+\omega_\ell}$ for some fixed scale factor $\tilde{\varsigma}_\ell > 0$ and exponent $\omega_\ell > 0$. Then, the limiting target-averaged generalization error is approximately*

$$
\bar{\epsilon} \simeq \begin{cases} \left(1 + \Omega_L + \sum_{\ell=1}^{L} \frac{1}{\alpha_\ell - 1}\right)\chi(\alpha_0) + \left(\omega_0 + \Omega_L + \sum_{\ell=0}^{L} \frac{1}{\alpha_\ell - 1}\right)\eta^2, & \alpha_0, \alpha_{\min} > 1 \\ \frac{\chi(\alpha_0/\alpha_{\min})}{1 - \alpha_{\min}} + \frac{\alpha_{\min}}{1 - \alpha_{\min}}\eta^2, & \alpha_{\min} < 1, \alpha_{\min} < \alpha_0 \\ \frac{\alpha_0}{1 - \alpha_0}\eta^2, & \alpha_0 < 1, \alpha_0 < \alpha_{\min}, \end{cases}
$$
(31)

*where $\Omega_L = \sum_{\ell=1}^{L} \omega_\ell$ and for $z > 1$ we have $\chi(z) \simeq -M_{\tilde{\Sigma}_0}^{-1}(z)/z$ given by $\chi(z) = \tilde{\varsigma}_0\left\{k(z^{\omega_0} - 1) + [2 + \omega_0(1 - k)](1 - 1/z)\right\}$ for $k = \text{sinc}[\pi/(1 + \omega_0)]^{-(1+\omega_0)}$.*

*Proof of Corollary 5.1.* Using the dictionary of notation in Appendix D, we can plug the approximate solutions for $\kappa_\ell$ and $\mu_\ell$ derived by Maloney et al. [19] into (27) to obtain (31). $\square$

Therefore, the power law exponents $\omega_1, \cdots, \omega_L$ of the weight covariances beyond the first layer, which enter only through their sum $\Omega_L$, do not affect the scaling laws of the generalization error with the dataset size and network widths. In particular, in the absence of label noise ($\eta = 0$) we can approximate the scaling of (31) in the regimes of large or small hidden layer width by

$$
\bar{\epsilon} \sim \begin{cases} \alpha_0^{\omega_0}, & \alpha_{\min} > 1, \alpha_0 \gg 1, \\ (\alpha_0/\alpha_{\min})^{\omega_0}, & \alpha_{\min} < 1, \alpha_0/\alpha_{\min} \gg 1, \end{cases}
$$
(32)

which recovers the results found by Maloney et al. [19] for $L = 1$ with unstructured weights. This behavior, and the agreement of (31) with numerical experiments, is illustrated in Figure 2. Consistent with Lemma 4.1, generalization with power-law weight structure is never better than with unstructured weights, as can be seen by comparing (31) with (25).

## 6 Bayesian inference and the Gibbs estimator at large prior variance

Thus far, we have focused on ridge regression (6). Though this is the most commonly-considered estimator in studies of random feature models [3–17, 19–21], one might ask whether our qualitative findings—in particular, that feature weight structure beyond the first layer is generally harmful for generalization—carry over to other estimators. Our approach to Proposition 3.2 is easily extensible to the setting of *zero-temperature Bayesian inference*, which has recently attracted substantial interest [13, 27, 34, 37, 45–47], sparked by work from Li and Sompolinsky [34]. In this case, we take seriously the Gibbs distribution $p(\mathbf{v}) \propto e^{-\beta L}$, which in the ridge regression case was simply a convenient tool, and interpret it as the Bayes posterior for a Gaussian likelihood of variance $1/\beta$ and a Gaussian prior with covariance $\Gamma_{L+1}/(\beta\lambda)$. It is in this context conventional to fix $\lambda = 1/\beta$, such that the prior variance does not scale with $\beta$. We can then study the average of the generalization error

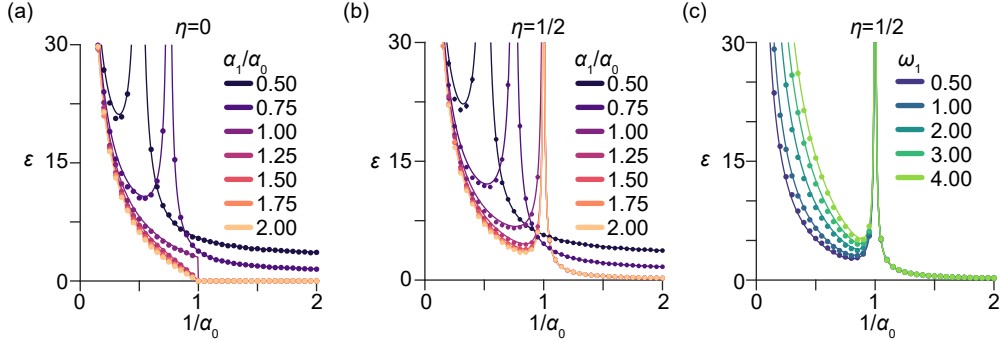

Figure 2: Generalization for power-law spectra. (a). Target-averaged generalization error $\bar{\epsilon}$ as a function of training data density $1/\alpha_0$ for shallow models ($L = 1$) of varying hidden layer width $\alpha_1/\alpha_0$ in the absence of label noise ($\eta = 0$). Here, the data and weight spectra have identical power law decay $\omega_0 = \omega_1 = 1$. (b). As in (a), but in the presence of label noise ($\eta = 1/2$). (c). As in (b), but for fixed hidden layer width $\alpha_1/\alpha_0 = 4$, fixed data exponent $\omega_0 = 1$, and varying weight exponents $\omega_1$. In all cases, solid lines show the predictions of (31), while dots with error bars show the mean and standard error over 100 realizations of numerical experiments with $n_0 = 1000$. See Appendix F for details of our numerical methods.

(13) under this posterior in the zero-temperature limit $\beta \to \infty$, which we refer to as the generalization error of the Gibbs estimator. We emphasize that this is not identical to the Bayesian minimum mean squared error (MMSE) estimator given by the posterior mean, which would coincide with the ridgeless estimator in the zero-temperature limit (see Appendix A).

For a deep RFM, this simply has the effect of adding a "thermal" variance term to the generalization error of the ridgeless estimator, which we describe in detail in Appendices A and C. We have:

**Proposition 6.1.** *With the same setup as in Proposition 3.2, the generalization error of the Gibbs estimator for a RFM is*

$$\epsilon_{\mathrm{BRF}} = \epsilon_{\mathrm{ridgeless}} + \begin{cases} \prod_{\ell=0}^{L} \frac{\kappa_\ell}{\alpha_\ell}, & \alpha_0, \alpha_{\min} > 1 \\ 0, & otherwise, \end{cases} \tag{33}$$

*where $\epsilon_{\mathrm{ridgeless}}$ is given by Proposition 3.2, and $\kappa_\ell$ is defined as in (20).*

*Proof of Proposition 6.1.* We derive (33) alongside Proposition 3.2 in Appendix A. □

The Gibbs estimator is sensitive to the scale of the random feature weight distributions through $\kappa_\ell$, while as noted above the ridgeless estimator is not sensitive to their overall scale. This direct dependence on $\kappa_\ell$ means that the simple argument of Lemma 4.1 cannot be applied. Indeed, in the limit of large prior variance, where the thermal variance term dominates, structure can improve the performance of the Gibbs estimator. We make this result precise in the following lemma:

**Lemma 6.1.** *In the setting of Proposition 6.1, consider Bayesian RFMs with weight covariances scaled as $\tau_\ell \tilde{\Sigma}_\ell$ for $\ell = 1, \ldots, L$. Then, in the non-trivial regime $\alpha_0, \alpha_{\min} > 1$ where the thermal variance is non-vanishing, we have*

$$\lim_{\tau_1, \ldots, \tau_L \to \infty} \frac{\epsilon_{\mathrm{BRF}}}{\prod_{\ell=1}^{L} \tau_\ell} = \prod_{\ell=0}^{L} \frac{\kappa_\ell}{\alpha_\ell} \leq \frac{\kappa_0}{\alpha_0} \varsigma^2 \prod_{\ell=1}^{L} \left(1 - \frac{1}{\alpha_\ell}\right), \tag{34}$$

*where the scalars $\kappa_\ell$ are defined in terms of the un-scaled covariances $\tilde{\Sigma}_\ell$ as in (20) and $\varsigma^2 \equiv \prod_{\ell=1}^{L} \mathbb{E}_{\tilde{\sigma}_\ell}[\tilde{\sigma}_\ell]$. Therefore, in the limit of large prior variance, including structure in the weight priors is generically advantageous for generalization. If $\mathbb{E}_{\tilde{\sigma}_\ell}[\tilde{\sigma}_\ell]$ is not finite, then the bound is vacuous.*

*Proof of Lemma 6.1.* The first part of (34) follows from (33) using the scaling properties of $\kappa_\ell$, while the bound follows from the bounds on $\kappa_\ell$ derived as part of Lemma 4.2. □

In contrast, weight structure is generally harmful for the Bayesian RFM in the limit of small prior variance, as its performance then coincides with the ridgeless RFM, as can be seen from the scaling

of $\kappa_\ell$. This example illustrates that there are cases in which, depending on the estimator used, weight structure in deeper layers can sometimes be helpful for generalization. However, whereas the ridgeless estimator is commonly used in practice, the Gibbs estimator is less standard, and the limit of large prior variance is certainly artificial. Therefore, we emphasize that we give this example to show that the behavior of the ridgeless estimator is not entirely general, not to show that weight structure can be helpful in practical settings.

## 7 Discussion

We have computed learning curves for models with many layers of structured Gaussian random features learning a linear target function, showing that structure beyond the first layer is generally detrimental for generalization. This result is consistent with the intuition that in deep linear models learning a single target direction it is sufficient to modify the representation only at the first layer [13, 36]. It will be interesting to investigate whether this intuition carries over to nonlinear networks learning complex tasks, particularly including multi-index targets [35, 48]. Moreover, we have considered only linear, Gaussian models. As mentioned in the Introduction, past works have established Gaussian equivalence theorems for nonlinear RFMs with unstructured Gaussian feature weights. It will be important to investigate the effect of feature weight structure on Gaussian equivalence in future work, and determine whether our qualitative results carry over to nonlinear RFMs in the proportional limit.

Though our results are obtained using the replica trick, and we do not address the possibility of replica symmetry breaking, they should be rigorously justifiable given the convexity of the ridge regression problem [29, 33, 41]. We note that the replica approach makes it straightforward to handle models of any finite depth [30]. The relevant averages could of course be computed with alternative random matrix theory techniques, which could allow for a fully rigorous proof [5, 19–21]. Another more challenging setting to study with either the replica trick or rigorous techniques would be that in which one allows for correlations between weights in different layers. This setting could qualitatively capture aspects of feature learning in deep networks, which induces couplings across depth [45].

In closing, we note that RFMs with structured weights may also have relevance for biological neural networks. A recent study by Pandey et al. [39] considered RFMs with a single layer of random features ($L = 1$) with correlated rows ($\mathbf{\Gamma}_1 \neq \mathbf{I}_{n_0}$). In several biologically-inspired settings, they showed that introducing this structure could improve generalization, consistent with our results. More broadly, biological neural networks are imbued with rich priors [49]; investigating what insights deep structured models can afford for neuroscience will be an interesting subject for further study.

## Acknowledgments and Disclosure of Funding

We thank Alexander Atanasov, Blake Bordelon, Benjamin S. Ruben, and James B. Simon for helpful discussions and comments on a draft of our manuscript. JAZ-V and CP were supported by NSF Award DMS-2134157 and NSF CAREER Award IIS-2239780. CP received additional support from a Sloan Research Fellowship. This work has been made possible in part by a gift from the Chan Zuckerberg Initiative Foundation to establish the Kempner Institute for the Study of Natural and Artificial Intelligence.

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

# A   Derivation of Proposition 3.2

In this Appendix, we sketch our replica-theory approach to computing the learning curves, which leads to Proposition 3.2. Many of the steps of this calculation are all but identical to our previous works on replica approaches to the spectra of product Wishart random matrices [30], and on unstructured deep Gaussian random feature models [13], so we will sketch the major steps rather than spelling out all the details of the algebra.

## A.1   Gibbs distribution and replica free energy

We start by introducing a Gibbs distribution at fictitious inverse temperature $\beta$ associated with the ridge regression loss

$$L = \frac{1}{2}\left\|\frac{1}{\sqrt{n_0}}\mathbf{X}\mathbf{F}\mathbf{v} - \mathbf{y}\right\|^2 + \frac{\lambda}{2}\|\mathbf{\Gamma}_{L+1}^{-1/2}\mathbf{v}\|_2^2, \tag{A.1}$$

with partition function

$$Z(\beta, \mathcal{D}) \propto \int d\mathbf{v}\, e^{-\beta L(\mathbf{v}, \mathcal{D})}, \tag{A.2}$$

where we denote by $\mathcal{D}$ all randomness in the problem. For any $\lambda > 0$, in the zero-temperature limit $\beta \to \infty$, this Gibbs distribution concentrates around the unique minimum of the loss $E$ [29, 33].

For the purpose of the replica computation, it is convenient to consider instead the partition function of the posterior of a related Bayesian model, which corresponds to absorbing $\beta\lambda$ into a redefinition of $\mathbf{\Gamma}_{L+1}$, and treating the ridge penalty as a Gaussian prior

$$\mathbf{v} \sim_{\text{prior}} \mathcal{N}(\mathbf{0}, \mathbf{\Gamma}_{L+1}). \tag{A.3}$$

We can then recover the partition function of the ridge regression model by undoing the rescaling: $\mathbf{\Gamma}_{L+1} \leftarrow \mathbf{\Gamma}_{L+1}/(\beta\lambda)$. Without this re-scaling—i.e., in the case in which the prior variance is held fixed as the temperature goes to zero—this is the Gibbs estimator in the zero-temperature limit, i.e., a Bayesian model with Gaussian likelihood of vanishing variance [13, 34, 37, 45, 46].

This gives us the partition function

$$Z = \mathbb{E}_{\mathbf{v}\sim\mathcal{N}(\mathbf{0},\mathbf{\Gamma}_{L+1})} \exp\left[-\frac{\beta}{2}\sum_{\mu=1}^{p}[g(\mathbf{x}_\mu; \mathbf{v}, \mathbf{F}) - y_\mu]^2\right], \tag{A.4}$$

which is the extension to structured priors of the Gibbs estimator partition function considered in [13]. By standard arguments, we expect the quenched free energy

$$f = -\lim_{p, n_0, \ldots, n_L \to \infty} \frac{1}{p}\log Z, \tag{A.5}$$

to be self-averaging in the thermodynamic limit, i.e., $f = \mathbb{E}_{\mathcal{D}}f$ almost surely [29, 33]. To compute the limiting quenched average, we use the replica trick, and write

$$f = -\lim_{m\to 0}\lim_{p, n_0, \ldots, n_L \to \infty} \frac{1}{pm}\log \mathbb{E}_{\mathcal{D}}Z^m, \tag{A.6}$$

where we evaluate the moments $\mathbb{E}_{\mathcal{D}}Z^m$ for positive integer $m$, and assume that they can be analytically continued to $m \to 0$.

Following previous work [13, 30], we can compute the quenched averages and integrate out the weights by introducing order parameters

$$(C_0)^{ab} = \frac{1}{n_0}(\mathbf{F}\mathbf{v}^a - \mathbf{w}_*)^\top\mathbf{\Sigma}_0(\mathbf{F}\mathbf{v}^b - \mathbf{w}_*), \tag{A.7}$$

for $\ell = 0$,

$$(C_\ell)^{ab} = \frac{1}{n_\ell \cdots n_L}(\mathbf{v}^a)^\top\mathbf{U}_L^\top\cdots\mathbf{U}_{\ell+1}^\top\mathbf{\Sigma}_\ell\mathbf{U}_{\ell+1}\cdots\mathbf{U}_L\mathbf{v}^b \tag{A.8}$$

for $\ell = 1, \dots, L - 1$ and

$$(C_L)^{ab} = \frac{1}{n_L}(\mathbf{v}^a)^\top \mathbf{\Sigma}_L \mathbf{v}^b, \tag{A.9}$$

along with corresponding Lagrange multipliers $\hat{\mathbf{C}}_\ell$, which yields

$$\mathbb{E}_\mathcal{D} Z^m = \int \frac{d\mathbf{C}_0 \, d\hat{\mathbf{C}}_0}{(4\pi i/n_0)^{m(m+1)/2}} \int \frac{d\mathbf{C}_1 \, d\hat{\mathbf{C}}_1}{(4\pi i/n_1)^{m(m+1)/2}} \cdots \int \frac{d\mathbf{C}_L \, d\hat{\mathbf{C}}_L}{(4\pi i/n_L)^{m(m+1)/2}} \exp\left[-\frac{pm}{2}S\right] \tag{A.10}$$

for

$$\begin{aligned}
mS = {} & \log \det(\mathbf{I}_m + \beta \mathbf{C}_0 + \beta\eta^2 \mathbf{1}_m \mathbf{1}_m^\top) \\
& - \alpha_0 \frac{1}{n_0} \mathrm{v}(\tilde{\mathbf{w}}_* \mathbf{1}_m^\top)^\top [\hat{\mathbf{C}}_0 \otimes \tilde{\mathbf{\Sigma}}_0][\mathbf{I}_{mn_0} - \mathbf{C}_1 \hat{\mathbf{C}}_0 \otimes \tilde{\mathbf{\Sigma}}_0]^{-1} \mathrm{v}(\tilde{\mathbf{w}}_* \mathbf{1}_m^\top) \\
& + \sum_{\ell=0}^{L} \alpha_\ell \left[ \mathrm{tr}(\mathbf{C}_\ell \hat{\mathbf{C}}_\ell) + \frac{1}{n_\ell} \log \det(\mathbf{I}_{mn_\ell} - \mathbf{C}_{\ell+1} \hat{\mathbf{C}}_\ell \otimes \tilde{\mathbf{\Sigma}}_\ell) \right], \tag{A.11}
\end{aligned}$$

where we let $\mathbf{C}_{L+1} = \mathbf{I}_m$ and

$$\tilde{\mathbf{\Sigma}}_\ell = \mathbf{\Gamma}_{\ell+1}^{1/2} \mathbf{\Sigma}_\ell \mathbf{\Gamma}_\ell^{1/2} \tag{A.12}$$

for $\ell = 0, \dots, L$. We note that $\otimes$ here denotes the Kronecker product, and we use the convention that the standard matrix product has higher precedence than the Kronecker product, i.e., $\mathbf{AB} \otimes \mathbf{C} = (\mathbf{AB}) \otimes \mathbf{C}$. Importantly, the quantity of interest—the generalization error—is simply given by the diagonal elements of $\mathbf{C}_0$, i.e., $\epsilon = (C_0)^{aa}$. Therefore, if we can solve for the order parameters at zero temperature, we will obtain the generalization error.

In the thermodynamic limit, the integral over these order parameters can be evaluated using the method of steepest descent. We make a replica symmetric *Ansatz*, and seek saddle points of the form

$$\mathbf{C}_\ell = q_\ell \mathbf{I}_m + c_\ell \mathbf{1}_m \mathbf{1}_m^\top, \tag{A.13}$$
$$\hat{\mathbf{C}}_\ell = \hat{q}_\ell \mathbf{I}_m + \hat{c}_\ell \mathbf{1}_m \mathbf{1}_m^\top. \tag{A.14}$$

Under this *Ansatz*, we have

$$\begin{aligned}
S = {} & \log(1 + \beta q_0) + \frac{\beta(c_0 + \eta^2)}{1 + \beta q_0} \\
& - \alpha_0 \frac{1}{n_0}(\tilde{\mathbf{w}}_*^\top \tilde{\mathbf{\Sigma}}_0 (\mathbf{I}_{n_0} - q_1 \hat{q}_0 \tilde{\mathbf{\Sigma}}_0)^{-1} \tilde{\mathbf{w}}_*) \hat{q}_0 \\
& + \sum_{\ell=0}^{L} \alpha_\ell \Bigg( q_\ell \hat{q}_\ell + q_\ell \hat{c}_\ell + c_\ell \hat{q}_\ell + \mathbb{E}_{\tilde{\sigma}_\ell} \log(1 - q_{\ell+1}\hat{q}_\ell \tilde{\sigma}_\ell) \\
& \qquad\qquad - (q_{\ell+1}\hat{c}_\ell + c_{\ell+1}\hat{q}_\ell)\mathbb{E}_{\tilde{\sigma}_\ell}\left[ \frac{\tilde{\sigma}_\ell}{1 - q_{\ell+1}\hat{q}_\ell \tilde{\sigma}_\ell} \right] \Bigg) \\
& + \mathcal{O}(m) \tag{A.15}
\end{aligned}$$

to leading order in $m$, where we recall the boundary condition $q_{L+1} = 1$, $c_{L+1} = 0$ [30]. The resulting saddle point equations can be simplified to give a closed system for the replica non-uniform components,

$$\alpha_0 \hat{q}_0 = -\frac{\beta}{1 + \beta q_0} \tag{A.16}$$

$$\alpha_\ell \hat{q}_\ell = \alpha_{\ell-1}\hat{q}_{\ell-1}\mathbb{E}_{\tilde{\sigma}_{\ell-1}}\left[ \frac{\tilde{\sigma}_{\ell-1}}{1 - q_\ell \hat{q}_{\ell-1}\tilde{\sigma}_{\ell-1}} \right] \qquad (\ell = 1, \dots, L) \tag{A.17}$$

$$q_\ell = q_{\ell+1}\mathbb{E}_{\tilde{\sigma}_\ell}\left[ \frac{\tilde{\sigma}_\ell}{1 - q_{\ell+1}\hat{q}_\ell \tilde{\sigma}_\ell} \right] \qquad (\ell = 0, \dots, L) \tag{A.18}$$

with the boundary condition $q_{L+1} = 1$, and a linear system for the replica uniform components,

$$\alpha_0 \hat{c}_0 = \frac{\beta^2(c_0 + \eta^2)}{(1 + \beta q_0)^2} \tag{A.19}$$

$$\alpha_1 \hat{c}_1 = \alpha_0 \frac{1}{n_0}(\tilde{\mathbf{w}}_*^\top \tilde{\boldsymbol{\Sigma}}_0^2 (\mathbf{I}_{n_0} - q_1 \hat{q}_0 \tilde{\boldsymbol{\Sigma}}_0)^{-2} \tilde{\mathbf{w}}_*) \hat{q}_0^2$$

$$+ \alpha_0 \left( \hat{c}_0 \mathbb{E}_{\tilde{\sigma}_0} \left[ \frac{\tilde{\sigma}_0}{1 - q_1 \hat{q}_0 \tilde{\sigma}_0} \right] + (q_1 \hat{c}_0 + c_1 \hat{q}_0) \hat{q}_0 \mathbb{E}_{\tilde{\sigma}_0} \left[ \left( \frac{\tilde{\sigma}_0}{1 - q_1 \hat{q}_0 \tilde{\sigma}_0} \right)^2 \right] \right) \tag{A.20}$$

$$\frac{\alpha_\ell}{\alpha_{\ell-1}} \hat{c}_\ell = \hat{c}_{\ell-1} \mathbb{E}_{\tilde{\sigma}_{\ell-1}} \left[ \frac{\tilde{\sigma}_{\ell-1}}{1 - q_\ell \hat{q}_{\ell-1} \tilde{\sigma}_{\ell-1}} \right]$$

$$+ (q_\ell \hat{c}_{\ell-1} + c_\ell \hat{q}_{\ell-1}) \hat{q}_{\ell-1} \mathbb{E}_{\tilde{\sigma}_{\ell-1}} \left[ \left( \frac{\tilde{\sigma}_{\ell-1}}{1 - q_\ell \hat{q}_{\ell-1} \tilde{\sigma}_{\ell-1}} \right)^2 \right] \qquad (\ell = 2, \ldots, L) \tag{A.21}$$

$$c_0 = \frac{1}{n_0}(\tilde{\mathbf{w}}_*^\top \tilde{\boldsymbol{\Sigma}}_0 (\mathbf{I}_{n_0} - q_1 \hat{q}_0 \tilde{\boldsymbol{\Sigma}}_0)^{-2} \tilde{\mathbf{w}}_*)$$

$$+ \left( c_1 \mathbb{E}_{\tilde{\sigma}_0} \left[ \frac{\tilde{\sigma}_0}{1 - q_1 \hat{q}_0 \tilde{\sigma}_0} \right] + (q_1 \hat{c}_0 + c_1 \hat{q}_0) q_1 \mathbb{E} \left[ \left( \frac{\tilde{\sigma}_0}{1 - q_1 \hat{q}_0 \tilde{\sigma}_0} \right)^2 \right] \right) \tag{A.22}$$

$$c_\ell = c_{\ell+1} \mathbb{E}_{\tilde{\sigma}_\ell} \left[ \frac{\tilde{\sigma}_\ell}{1 - q_{\ell+1} \hat{q}_\ell \tilde{\sigma}_\ell} \right]$$

$$+ (q_{\ell+1} \hat{c}_\ell + c_{\ell+1} \hat{q}_\ell) q_{\ell+1} \mathbb{E}_{\tilde{\sigma}_\ell} \left[ \left( \frac{\tilde{\sigma}_\ell}{1 - q_{\ell+1} \hat{q}_\ell \tilde{\sigma}_\ell} \right)^2 \right] \qquad (\ell = 1, \ldots, L) \tag{A.23}$$

with the boundary condition $c_{L+1} = 0$.

### A.2 Converting between the Gibbs and maximum-likelihood estimators

As our primary aim is to study ridge regression, we must now account for the fact that the prior over the readout weights scales with the inverse temperature $\beta$. In particular, we have a prior with scaled covariance $\boldsymbol{\Gamma}_{L+1}/(\beta\lambda)$, where $\boldsymbol{\Gamma}_{L+1}$ does not scale with $\beta$. If we perform this rescaling in (A.16) and (A.16), we can see that the re-scaled order parameters

$$\bar{q}_\ell = \beta\lambda q_\ell \tag{A.24}$$

$$\bar{\hat{q}}_\ell = \frac{1}{\beta\lambda} \hat{q}_\ell \tag{A.25}$$

$$\bar{c}_\ell = c_\ell \tag{A.26}$$

$$\bar{\hat{c}}_\ell = \frac{1}{(\beta\lambda)^2} \hat{c}_\ell \tag{A.27}$$

obey an identical system of equations to the original order parameters in the Bayesian case at inverse temperature

$$\beta = \frac{1}{\lambda}. \tag{A.28}$$

Therefore, if we can solve the saddle point equations for the Gibbs estimator in the zero-temperature limit, we can simply read off the corresponding result for the ridge regression estimator in the ridgeless limit. The important difference is that the replica nonuniform component $q_0$ of $\mathbf{C}_0$ is $\mathcal{O}(1/\beta)$ in the ridge regression case, hence only the replica uniform component $c_0$ contributes to the generalization error. We note that this allows one to read off the generalization error of a deep linear RFM with unstructured features from the results of our previous work [13] simply by setting the Bayesian prior variance $\sigma^2$ to zero.

### A.3 Solutions for the generalization error

The replica-symmetric saddle point equations in (A.16) and (A.19) are nearly identical to those analyzed our computation of the maximum eigenvalue of a structured Wishart product matrix [30],

which in turn are related to those in our original paper on unstructured deep linear RFMs [13] by the replacement of the spectral moment generating function of the identity matrix with the appropriate spectral generating functions. Given this similarity, and the fact that we have provided extensive exposition of how to solve such systems in those previous works, we will merely state the results for the order parameters relevant to the computation of the generalization error.

Let

$$M_{\tilde{\boldsymbol{\Sigma}}_\ell}(z) = \lim_{n_\ell \to \infty} \frac{1}{n_\ell} \text{tr}[\tilde{\boldsymbol{\Sigma}}_\ell (z\mathbf{I}_{n_\ell} - \tilde{\boldsymbol{\Sigma}}_\ell)^{-1}] \tag{A.29}$$

be the moment generating function of $\tilde{\boldsymbol{\Sigma}}_\ell$, with functional inverse $M_{\tilde{\boldsymbol{\Sigma}}_\ell}^{-1}(z)$. Then, at finite temperature, after eliminating the Lagrange multipliers, the replica nonuniform components of the order parameters are given by

$$q_\ell = \prod_{j=\ell}^{L} \frac{A}{\alpha_j} M_{\tilde{\boldsymbol{\Sigma}}_j}^{-1}\left(\frac{A}{\alpha_j}\right) \tag{A.30}$$

for $\ell = 0, \ldots, L$, where

$$A = q_0 \hat{q}_0 = -\frac{\beta q_0}{1 + \beta q_0} \tag{A.31}$$

satisfies the closed equation

$$-\frac{1}{\beta} = \frac{A+1}{A} \prod_{\ell=0}^{L} \frac{A}{\alpha_\ell} M_{\tilde{\boldsymbol{\Sigma}}_\ell}^{-1}\left(\frac{A}{\alpha_\ell}\right). \tag{A.32}$$

From [30], we recognize this as the self-consistent equation for the moment generating function $M = A$ of the feature kernel $\mathbf{K} = \mathbf{X}\mathbf{F}\mathbf{F}^\top\mathbf{X}^\top/n_0$, evaluated at $-1/\beta$. Even in the unstructured case, this equation must in general be solved numerically at finite temperature [30].

Given a solution to this equation, we can solve the system of linear equations (A.19) for $c_0$, mirroring the computation of the extremal eigenvalues of structured product Wishart matrices in [30]. After eliminating the Lagrange multipliers, this calculation boils down to solving a three-term recurrence relation, which is detailed in previous works [13, 30]. We therefore simply state the result of this computation here. Let $\zeta = -A$, which then satisfies

$$\lambda = \frac{1-\zeta}{\zeta} \prod_{\ell=0}^{L} \frac{-\zeta}{\alpha_\ell} M_{\tilde{\boldsymbol{\Sigma}}_\ell}^{-1}\left(-\frac{\zeta}{\alpha_\ell}\right). \tag{A.33}$$

For $\ell = 0, \ldots, L$, let

$$\kappa_\ell = -M_{\tilde{\boldsymbol{\Sigma}}_\ell}^{-1}\left(-\frac{\zeta}{\alpha_\ell}\right) \tag{A.34}$$

so that $\kappa_\ell$ satisfies

$$\frac{\zeta}{\alpha_\ell} = \mathbb{E}_{\tilde{\sigma}_\ell}\left[\frac{\tilde{\sigma}_\ell}{\kappa_\ell + \tilde{\sigma}_\ell}\right]. \tag{A.35}$$

Viewing $\kappa_\ell$ as a function of $\zeta$, we may alternatively write the self-consistent equation for $\zeta$ as

$$\frac{1}{\beta} = \frac{1-\zeta}{\zeta} \prod_{\ell=0}^{L} \frac{\zeta}{\alpha_\ell} \kappa_\ell(\zeta) \tag{A.36}$$

In terms of $\kappa_\ell$, let

$$\mu_\ell = -\frac{\alpha_\ell}{\zeta} \kappa_\ell M_{\tilde{\boldsymbol{\Sigma}}_\ell}'\left(-\kappa_\ell\right) \tag{A.37}$$

$$= 1 - \frac{\alpha_\ell}{\zeta} \mathbb{E}_{\tilde{\sigma}_\ell}\left[\left(\frac{\tilde{\sigma}_\ell}{\kappa_\ell + \tilde{\sigma}_\ell}\right)^2\right] \tag{A.38}$$

We then finally have

$$\left[1 + \left(\sum_{\ell=0}^{L} \frac{1-\mu_\ell}{\mu_\ell}\right)(1-\zeta)\right] c_0 = \frac{1}{\mu_0}\frac{1}{n_0}(\tilde{\mathbf{w}}_*^\top \tilde{\mathbf{\Sigma}}_0(\kappa_0 \mathbf{I}_{n_0} + \tilde{\mathbf{\Sigma}}_0)^{-2}\tilde{\mathbf{w}}_*)\kappa_0^2$$

$$+ \left(\sum_{\ell=1}^{L} \frac{1-\mu_\ell}{\mu_\ell}\right)\frac{1}{n_0}(\tilde{\mathbf{w}}_*^\top \tilde{\mathbf{\Sigma}}_0(\kappa_0 \mathbf{I}_{n_0} + \tilde{\mathbf{\Sigma}}_0)^{-1}\tilde{\mathbf{w}}_*)\kappa_0$$

$$+ \left(\sum_{\ell=0}^{L} \frac{1-\mu_\ell}{\mu_\ell}\right)\zeta\eta^2. \tag{A.39}$$

Using the mapping of Appendix A.2 and again defining the weighted generating function

$$\psi(z) = \lim_{n_0 \to \infty} \frac{1}{n_0}\tilde{\mathbf{w}}_*^\top \tilde{\mathbf{\Sigma}}_0(z\mathbf{I}_{n_0} + \tilde{\mathbf{\Sigma}}_0)^{-1}\tilde{\mathbf{w}}_*. \tag{A.40}$$

this yields Proposition 3.1.

We now want to extract the zero-temperature/ridgeless limit. As $\beta \to \infty$, the self-consistent equation for $\zeta$ admits the solution

$$\zeta = 1, \tag{A.41}$$

valid for $\alpha_\ell > 1$ for all $\ell$, which gives $q_0 \sim \mathcal{O}(1)$, the solution

$$\zeta = \alpha_0, \tag{A.42}$$

valid for $\alpha_0 < 1$, $\alpha_0 < \alpha_1, \dots, \alpha_L$, which gives $q_0 \sim \mathcal{O}(1/\beta)$, and, for $\ell_* = 0, \dots, L$, the solutions

$$\zeta = \alpha_{\ell_*}, \tag{A.43}$$

valid for $\alpha_{\ell_*} < 1$, $\alpha_{\ell_*} < \alpha_0$, $\alpha_{\ell_*} < \alpha_\ell$ for all $\ell \neq \ell_*$, which also give $q_0 \sim \mathcal{O}(1/\beta)$. These solutions mirror those found in the unstructured setting [13]. We remark that, as in [13], we can determine the regimes in which each solution is physical by demanding that the order parameters $q_\ell$ are non-negative.

For the $\zeta \to 1$ solution, we immediately have

$$c_0 = \frac{1}{\mu_0}\frac{1}{n_0}(\tilde{\mathbf{w}}_*^\top \tilde{\mathbf{\Sigma}}_0(\kappa_0 \mathbf{I}_{n_0} + \tilde{\mathbf{\Sigma}}_0)^{-2}\tilde{\mathbf{w}}_*)\kappa_0^2$$

$$+ \left(\sum_{\ell=1}^{L} \frac{1-\mu_\ell}{\mu_\ell}\right)\frac{1}{n_0}(\tilde{\mathbf{w}}_*^\top \tilde{\mathbf{\Sigma}}_0(\kappa_0 \mathbf{I}_{n_0} + \tilde{\mathbf{\Sigma}}_0)^{-1}\tilde{\mathbf{w}}_*)\kappa_0$$

$$+ \left(\sum_{\ell=0}^{L} \frac{1-\mu_\ell}{\mu_\ell}\right)\zeta\eta^2, \tag{A.44}$$

where by a minor abuse of notation we simply write $\kappa_\ell$ and $\mu_\ell$ for the corresponding quantities evaluated at $\zeta = 1$.

If $\zeta \to \alpha_\ell$, then $\kappa_\ell \downarrow 0$ and $\mu_\ell \downarrow 0$. We can then apply L'Hôpital's rule to evaluate the limit in A.39, which corresponds to extracting the most divergent terms on each side of A.39. For the $\zeta = \alpha_0$ solution, one finds that

$$c_0 = \frac{\alpha_0}{1-\alpha_0}\eta^2. \tag{A.45}$$

Finally, for the solutions with $\zeta = \alpha_{\ell_*}$ for $\ell_* = 1, \dots, L$, one finds that

$$c_0 = \frac{1}{1-\alpha_{\ell_*}}\frac{1}{n_0}(\tilde{\mathbf{w}}_*^\top \tilde{\mathbf{\Sigma}}_0(\kappa_0 \mathbf{I}_{n_0} + \tilde{\mathbf{\Sigma}}_0)^{-1}\tilde{\mathbf{w}}_*)\kappa_0$$

$$+ \frac{\alpha_{\ell_*}}{1-\alpha_{\ell_*}}\eta^2, \tag{A.46}$$

where we must be careful to recall that $\kappa_0$ now satisfies

$$\frac{\alpha_{\ell_*}}{\alpha_0} = \mathbb{E}_{\tilde{\sigma}_0}\left[\frac{\tilde{\sigma}_0}{\kappa_0 + \tilde{\sigma}_0}\right]. \tag{A.47}$$

But, we recognize that $\alpha_{\ell_*} = \alpha_{\min} = \min\{\alpha_1, \ldots, \alpha_L\}$, so we will write

$$\kappa_{\min} = \kappa_0 \Big|_{\zeta = \alpha_{\min}} \tag{A.48}$$

to avoid clashing with our notation for the $\zeta = 1$ solution.

Therefore, recalling from Appendix A.2 that the generalization error for the ridge regression estimator in the ridgeless limit is simply given by $c_0$, we have

$$\epsilon_{\text{ridgeless}} = \begin{cases} \left(\sum_{\ell=1}^{L} \frac{1-\mu_\ell}{\mu_\ell}\right)\kappa_0\psi(\kappa_0) - \frac{\kappa_0^2}{\mu_0}\psi'(\kappa_0) + \left(\sum_{\ell=0}^{L} \frac{1-\mu_\ell}{\mu_\ell}\right)\eta^2, & \alpha_0, \alpha_{\min} > 1 \\ \frac{\kappa_{\min}\psi(\kappa_{\min})}{1-\alpha_{\min}} + \frac{\alpha_{\min}}{1-\alpha_{\min}}\eta^2, & \alpha_{\min} < 1, \alpha_{\min} < \alpha_0 \\ \frac{\alpha_0}{1-\alpha_0}\eta^2, & \alpha_0 < 1, \alpha_0 < \alpha_{\min}, \end{cases} \tag{A.49}$$

as reported in (23), where we again define the weighted generating function

$$\psi(z) = \lim_{n_0 \to \infty} \frac{1}{n_0} \tilde{\mathbf{w}}_*^\top \tilde{\mathbf{\Sigma}}_0 (z\mathbf{I}_{n_0} + \tilde{\mathbf{\Sigma}}_0)^{-1} \tilde{\mathbf{w}}_*. \tag{A.50}$$

To obtain the average generalization error for the Gibbs estimator in the zero-temperature limit, we must account for the effect of $q_0$ in the regime $\alpha_\ell > 1$, as in all other regimes it is $q_0 \sim \mathcal{O}(1/\beta)$. But, we recognize that

$$q_0 = \prod_{j=0}^{L} \frac{-1}{\alpha_j} M_{\tilde{\mathbf{\Sigma}}_j}^{-1}\left(\frac{-1}{\alpha_j}\right) = \prod_{\ell=0}^{L} \frac{\kappa_\ell}{\alpha_\ell} \tag{A.51}$$

from the definition above, hence we conclude that

$$\epsilon_{\text{BRFM}} = \epsilon_{\text{ridgeless}} + \begin{cases} \prod_{\ell=0}^{L} \frac{\kappa_\ell}{\alpha_\ell}, & \alpha_0, \alpha_{\min} > 1 \\ 0, & \alpha_{\min} < 1, \alpha_{\min} < \alpha_0 \\ 0, & \alpha_0 < 1, \alpha_0 < \alpha_{\min}. \end{cases} \tag{A.52}$$

### A.4 Physical interpretation of the order parameters and thermal bias-variance decomposition

With these results in hand, we now comment on the interpretation of the replica uniform and replica non-uniform contributions to

$$\mathbf{C}_0 = q_0\mathbf{I}_m + c_0\mathbf{1}_m\mathbf{1}_m^\top. \tag{A.53}$$

At the saddle point, we have

$$(C_0)^{ab} = \mathbb{E}_{\mathcal{D}}\left\langle \frac{1}{n_0}(\mathbf{F}\mathbf{v}^a - \mathbf{w}_*)^\top\mathbf{\Sigma}_0(\mathbf{F}\mathbf{v}^b - \mathbf{w}_*)\right\rangle_\beta, \tag{A.54}$$

where $\langle\cdot\rangle_\beta$ denotes the expectation with respect to the replicated Gibbs measure at inverse temperature $\beta$. Under the replica-symmetric *Ansatz*, considering off-diagonal elements $a \neq b$, we can use the fact that the replicas are initially uncoupled and identical to write

$$c_0 = C_0^{ab} \tag{A.55}$$

$$= \mathbb{E}_{\mathcal{D}}\frac{1}{n_0}(\mathbf{F}\langle\mathbf{v}^a\rangle_\beta - \mathbf{w}_*)^\top\mathbf{\Sigma}_0(\mathbf{F}\langle\mathbf{v}^b\rangle_\beta - \mathbf{w}_*) \tag{A.56}$$

$$= \mathbb{E}_{\mathcal{D}}\frac{1}{n_0}(\mathbf{F}\langle\mathbf{v}\rangle_\beta - \mathbf{w}_*)^\top\mathbf{\Sigma}_0(\mathbf{F}\langle\mathbf{v}\rangle_\beta - \mathbf{w}_*) \tag{A.57}$$

$$= \mathbb{E}_{\mathcal{D}}\frac{1}{n_0}\|\mathbf{\Sigma}_0^{1/2}(\mathbf{F}\langle\mathbf{v}\rangle_\beta - \mathbf{w}_*)\|^2. \tag{A.58}$$

Similarly, we have

$$q_0 = C_0^{aa} - C_0^{ab} \tag{A.59}$$

$$= \mathbb{E}_{\mathcal{D}} \left\langle \frac{1}{n_0} (\mathbf{F}\mathbf{v}^a - \mathbf{w}_*)^\top \mathbf{\Sigma}_0 (\mathbf{F}\mathbf{v}^a - \mathbf{w}_*) \right\rangle_\beta - c_0 \tag{A.60}$$

$$= \mathbb{E}_{\mathcal{D}} \left\langle \frac{1}{n_0} (\mathbf{F}\delta\mathbf{v} + \mathbf{F}\langle\mathbf{v}\rangle_\beta - \mathbf{w}_*)^\top \mathbf{\Sigma}_0 (\mathbf{F}\delta\mathbf{v} + \mathbf{F}\langle\mathbf{v}\rangle_\beta - \mathbf{w}_*) \right\rangle_\beta - c_0 \tag{A.61}$$

$$= \mathbb{E}_{\mathcal{D}} \left\langle \frac{1}{n_0} (\mathbf{F}\delta\mathbf{v})^\top \mathbf{\Sigma}_0 (\mathbf{F}\delta\mathbf{v}) \right\rangle_\beta + \mathbb{E}_{\mathcal{D}} \frac{1}{n_0} (\mathbf{F}\langle\mathbf{v}\rangle_\beta - \mathbf{w}_*)^\top \mathbf{\Sigma}_0 (\mathbf{F}\langle\mathbf{v}\rangle_\beta - \mathbf{w}_*) - c_0 \tag{A.62}$$

$$= \mathbb{E}_{\mathcal{D}} \left\langle \frac{1}{n_0} (\mathbf{F}\delta\mathbf{v})^\top \mathbf{\Sigma}_0 (\mathbf{F}\delta\mathbf{v}) \right\rangle_\beta \tag{A.63}$$

$$= \mathbb{E}_{\mathcal{D}} \left\langle \frac{1}{n_0} \|\mathbf{\Sigma}_0^{1/2} \mathbf{F}\delta\mathbf{v}\|^2 \right\rangle_\beta, \tag{A.64}$$

where we write $\delta\mathbf{v} = \mathbf{v} - \langle\mathbf{v}\rangle_\beta$. Therefore, at the saddle point, $c_0$ and $q_0$ correspond exactly to the bias and variance terms in the thermal bias-variance decomposition of the generalization error:

$$\mathbb{E}_{\mathcal{D}} \left\langle \frac{1}{n_0} \|\mathbf{\Sigma}_0^{1/2} (\mathbf{F}\mathbf{v} - \mathbf{w}_*)\|^2 \right\rangle_\beta = \mathbb{E}_{\mathcal{D}} \frac{1}{n_0} \|\mathbf{\Sigma}_0^{1/2} (\mathbf{F}\langle\mathbf{v}\rangle_\beta - \mathbf{w}_*)\|^2 + \mathbb{E}_{\mathcal{D}} \left\langle \frac{1}{n_0} \|\mathbf{\Sigma}_0^{1/2} \mathbf{F}\delta\mathbf{v}\|^2 \right\rangle_\beta.$$
$$\tag{A.65}$$

This makes concrete an argument which was presented only intuitively in [13]. As a result, if one considered the Bayesian MMSE estimator $\hat{\mathbf{v}} = \langle\mathbf{v}\rangle_\beta$, the zero-temperature generalization error would simply coincide with that for the ridgeless estimator.

# B  Properties of the inverse generating functions

Here, we record a few useful properties of the inverse spectral generating functions

$$\frac{1}{\alpha_\ell} = -M_{\tilde{\mathbf{\Sigma}}_\ell}(-\kappa_\ell) = \mathbb{E}_{\tilde{\sigma}_\ell} \left[ \frac{\tilde{\sigma}_\ell}{\kappa_\ell + \tilde{\sigma}_\ell} \right] \tag{B.1}$$

and their relatives

$$\mu_\ell = -\alpha_\ell \kappa_\ell M'_{\tilde{\mathbf{\Sigma}}_\ell}(-\kappa_\ell) = 1 - \alpha_\ell \mathbb{E}_{\tilde{\sigma}_\ell} \left[ \left( \frac{\tilde{\sigma}_\ell}{\kappa_\ell + \tilde{\sigma}_\ell} \right)^2 \right]. \tag{B.2}$$

These results are used in the proofs of Lemmas

## B.1  Dependence on width

Implicitly differentiating the self-consistent equation defining $\kappa_\ell$, we have

$$\frac{d\kappa_\ell}{d(1/\alpha_\ell)} = -\frac{1}{\mathbb{E}_{\tilde{\sigma}_\ell} \left[ \frac{\tilde{\sigma}_\ell}{(\kappa_\ell + \tilde{\sigma}_\ell)^2} \right]}, \tag{B.3}$$

showing that $\kappa_\ell$ is a decreasing function of $1/\alpha_\ell$. As $1/\alpha_\ell \downarrow 0$, we should have $\kappa_\ell \uparrow \infty$, while as $1/\alpha_\ell \uparrow 1$, we should have $\kappa_\ell \downarrow 0$.

## B.2  Behavior under rescaling

Consider the re-scaling $\tilde{\mathbf{\Sigma}}'_\ell = \tau_\ell \tilde{\mathbf{\Sigma}}_\ell$ for $\tau_\ell > 0$. Then, we have $\kappa_\ell$ and $\tilde{\kappa}'_\ell$ given by

$$\frac{1}{\alpha_\ell} = -M_{\tilde{\mathbf{\Sigma}}_\ell}(-\kappa_\ell) = \mathbb{E}_{\tilde{\sigma}_\ell} \left[ \frac{\tilde{\sigma}_\ell}{\kappa_\ell + \tilde{\sigma}_\ell} \right] \tag{B.4}$$

and

$$\frac{1}{\alpha_\ell} = -M_{\tilde{\mathbf{\Sigma}}'_\ell}(-\kappa'_\ell) = \mathbb{E}_{\tilde{\sigma}_\ell} \left[ \frac{\tau_\ell \tilde{\sigma}_\ell}{\kappa'_\ell + \tau_\ell \tilde{\sigma}_\ell} \right] \tag{B.5}$$

respectively. We can then see that we should have

$$\kappa'_\ell = \tau_\ell \kappa_\ell. \tag{B.6}$$

### B.3 Bound on $\kappa_\ell$ in terms of isotropic spectrum

We now prove that

$$\kappa_\ell \leq (\alpha_\ell - 1)\mathbb{E}[\tilde{\sigma}_\ell] \tag{B.7}$$

in the relevant regime $\alpha_\ell > 1$. For any $z > 0$,

$$\tilde{\sigma}_\ell \mapsto \frac{\tilde{\sigma}_\ell}{(z + \tilde{\sigma}_\ell)} \tag{B.8}$$

is a concave function of $\tilde{\sigma}_\ell \geq 0$, hence Jensen's inequality implies that

$$\mathbb{E}_{\tilde{\sigma}_\ell}\left[\frac{\tilde{\sigma}_\ell}{(z + \tilde{\sigma}_\ell)}\right] \leq \frac{\mathbb{E}[\tilde{\sigma}_\ell]}{z + \mathbb{E}[\tilde{\sigma}_\ell]}. \tag{B.9}$$

Then, note that

$$z \mapsto \mathbb{E}_{\tilde{\sigma}_\ell}\left[\frac{\tilde{\sigma}_\ell}{z + \tilde{\sigma}_\ell}\right] \tag{B.10}$$

and

$$z \mapsto \frac{\mathbb{E}[\tilde{\sigma}_\ell]}{z + \mathbb{E}[\tilde{\sigma}_\ell]} \tag{B.11}$$

are both decreasing functions of $z \geq 0$, and both are equal to 1 when $z = 0$. Thus, if $\kappa_\ell > 0$ solves

$$\frac{1}{\alpha_\ell} = \mathbb{E}_{\tilde{\sigma}_\ell}\left[\frac{\tilde{\sigma}_\ell}{\kappa_\ell + \tilde{\sigma}_\ell}\right] \tag{B.12}$$

as specified by its definition and $\bar{\kappa}_\ell > 0$ solves

$$\frac{1}{\alpha_\ell} = \frac{\mathbb{E}[\tilde{\sigma}_\ell]}{\bar{\kappa}_\ell + \mathbb{E}[\tilde{\sigma}_\ell]}, \tag{B.13}$$

we must have

$$\kappa_\ell \leq \bar{\kappa}_\ell. \tag{B.14}$$

But, we can easily see that $\bar{\kappa}_\ell = (\alpha_\ell - 1)\mathbb{E}[\tilde{\sigma}_\ell]$, hence the claim follows.

### B.4 Bound on $\mu_\ell$ terms of isotropic spectrum

We next prove that

$$\mu_\ell \leq 1 - \frac{1}{\alpha_\ell} \tag{B.15}$$

in the relevant regime $\alpha_\ell > 1$. By definition, we have

$$\mu_\ell = 1 - \alpha_\ell \mathbb{E}_{\tilde{\sigma}_\ell}\left[\left(\frac{\tilde{\sigma}_\ell}{\kappa_\ell + \tilde{\sigma}_\ell}\right)^2\right]. \tag{B.16}$$

By Jensen's inequality and the definition of $\kappa_\ell$, we have

$$\mathbb{E}_{\tilde{\sigma}_\ell}\left[\left(\frac{\tilde{\sigma}_\ell}{\kappa_\ell + \tilde{\sigma}_\ell}\right)^2\right] \geq \mathbb{E}_{\tilde{\sigma}_\ell}\left[\frac{\tilde{\sigma}_\ell}{\kappa_\ell + \tilde{\sigma}_\ell}\right]^2 \tag{B.17}$$

$$= \frac{1}{\alpha_\ell^2}. \tag{B.18}$$

As $\alpha_\ell > 1$ by assumption, this bound is always positive. Therefore, we conclude the desired claim.

## C   Simplifying the generalization error for fixed data

In this appendix, we show how the ridgeless generalization error can be simplified in each regime for fixed data. Using the solution to the ridge regression problem (6),

$$\hat{\mathbf{v}} = \frac{1}{\sqrt{n_0}} \left( \lambda \mathbf{\Gamma}_{L+1}^{-1} + \frac{1}{n_0} \mathbf{F}^\top \mathbf{X}^\top \mathbf{X} \mathbf{F} \right)^{-1} \mathbf{F}^\top \mathbf{X}^\top \mathbf{y}, \tag{C.1}$$

we have

$$\epsilon = \lim_{\lambda \downarrow 0} \lim_{p, n_0, \dots, n_L \to \infty} \mathbb{E}_{\mathcal{D}} \frac{1}{n_0} \| \mathbf{\Sigma}_0^{1/2} (\mathbf{F}\hat{\mathbf{v}} - \mathbf{w}_*) \|^2 \tag{C.2}$$

$$= \lim_{\lambda \downarrow 0} \lim_{p, n_0, \dots, n_L \to \infty} \mathbb{E}_{\mathcal{D}} \frac{1}{n_0} \left\| \frac{1}{\sqrt{n_0}} \mathbf{\Sigma}_0^{1/2} \mathbf{F} \left( \lambda \mathbf{\Gamma}_{L+1}^{-1} + \frac{1}{n_0} \mathbf{F}^\top \mathbf{X}^\top \mathbf{X} \mathbf{F} \right)^{-1} \mathbf{F}^\top \mathbf{X}^\top \mathbf{y} - \mathbf{\Sigma}_0^{1/2} \mathbf{w}_* \right\|^2. \tag{C.3}$$

Following our discussion in the main text, we may set $\mathbf{\Gamma}_{L+1} = \mathbf{I}_{n_L}$ without loss of generality, as otherwise we may re-define $\mathbf{\Sigma}_L$. Then, we have

$$\epsilon = \lim_{\lambda \downarrow 0} \lim_{p, n_0, \dots, n_L \to \infty} \mathbb{E}_{\mathcal{D}} \frac{1}{n_0} \left\| \frac{1}{\sqrt{n_0}} \mathbf{\Sigma}_0^{1/2} \mathbf{F} \left( \lambda \mathbf{I}_{n_L} + \frac{1}{n_0} \mathbf{F}^\top \mathbf{X}^\top \mathbf{X} \mathbf{F} \right)^{-1} \mathbf{F}^\top \mathbf{X}^\top \mathbf{y} - \mathbf{\Sigma}_0^{1/2} \mathbf{w}_* \right\|^2. \tag{C.4}$$

In the subsequent sections, we will simplify this expression in each regime.

For the Gibbs estimator, we must account for the additional contribution to the generalization error from thermal variance. Following our previous work [13], we may compute the bias and variance terms directly from the posterior moment generating function of the readout weight vector,

$$\mathcal{Z}(\mathbf{j}) \propto \int d\mathbf{v} \, \exp \left( -\frac{\beta}{2} \| n_0^{-1/2} \mathbf{X}\mathbf{F}\mathbf{v} - \mathbf{y} \|^2 - \frac{1}{2} \| \mathbf{\Gamma}_{L+1}^{-1/2} \mathbf{v} \|^2 + \mathbf{j}^\top \mathbf{v} \right) \tag{C.5}$$

$$\propto \exp \left( \beta n_0^{-1/2} \mathbf{y}^\top \mathbf{X}\mathbf{F}(\mathbf{\Gamma}_{L+1}^{-1} + \beta n_0^{-1} \mathbf{F}^\top \mathbf{X}^\top \mathbf{X}\mathbf{F})^{-1} \mathbf{j} \right.$$

$$\left. + \frac{1}{2} \mathbf{j}^\top (\mathbf{\Gamma}_{L+1}^{-1} + \beta n_0^{-1} \mathbf{F}^\top \mathbf{X}^\top \mathbf{X}\mathbf{F})^{-1} \mathbf{j} \right), \tag{C.6}$$

yielding

$$\langle \mathbf{v} \rangle_\beta = \frac{1}{\sqrt{n_0}} \left( \frac{1}{\beta} \mathbf{\Gamma}_{L+1}^{-1} + \frac{1}{n_0} \mathbf{F}^\top \mathbf{X}^\top \mathbf{X} \mathbf{F} \right)^{-1} \mathbf{F}^\top \mathbf{X}^\top \mathbf{y} \tag{C.7}$$

and

$$\langle \mathbf{v}\mathbf{v}^\top \rangle_\beta - \langle \mathbf{v} \rangle_\beta \langle \mathbf{v} \rangle_\beta^\top = \left( \mathbf{\Gamma}_{L+1}^{-1} + \frac{\beta}{n_0} \mathbf{F}^\top \mathbf{X}^\top \mathbf{X} \mathbf{F} \right)^{-1}. \tag{C.8}$$

We then can see that

$$\langle \mathbf{v} \rangle_\beta = \hat{\mathbf{v}} \Big|_{\lambda = 1/\beta}, \tag{C.9}$$

which is precisely in agreement with the conversion in Appendix A.2. Considering the thermal bias-variance decomposition of the generalization error for the Gibbs estimator,

$$\mathbb{E}_{\mathcal{D}} \left\langle \frac{1}{n_0} \| \mathbf{\Sigma}_0^{1/2} (\mathbf{F}\mathbf{v} - \mathbf{w}_*) \|^2 \right\rangle_\beta = \mathbb{E}_{\mathcal{D}} \frac{1}{n_0} \| \mathbf{\Sigma}_0^{1/2} (\mathbf{F}\langle \mathbf{v} \rangle_\beta - \mathbf{w}_*) \|^2 + \mathbb{E}_{\mathcal{D}} \left\langle \frac{1}{n_0} \| \mathbf{\Sigma}_0^{1/2} \mathbf{F} \delta\mathbf{v} \|^2 \right\rangle_\beta, \tag{C.10}$$

we can then see that the bias term at zero temperature coincides exactly with the generalization error of the ridgeless estimator, as we found in Appendix A. The variance term is

$$\lim_{\beta \to \infty} \mathbb{E}_{\mathcal{D}} \left\langle \frac{1}{n_0} \| \mathbf{\Sigma}_0^{1/2} \mathbf{F} \delta\mathbf{v} \|^2 \right\rangle_\beta = \lim_{\beta \to \infty} \mathbb{E}_{\mathcal{D}} \frac{1}{n_0} \, \text{tr} \left[ \mathbf{\Sigma}_0 \mathbf{F} \left( \mathbf{\Gamma}_{L+1}^{-1} + \frac{\beta}{n_0} \mathbf{F}^\top \mathbf{X}^\top \mathbf{X} \mathbf{F} \right)^{-1} \mathbf{F}^\top \right]. \tag{C.11}$$

In both the bias and variance terms, we can see that we may set $\mathbf{\Gamma}_{L+1} = \mathbf{I}_{n_L}$ without loss of generality, as otherwise we may simply re-scale $\mathbf{\Sigma}_L$ as discussed in Lemma 2.1. Then, we need only consider the thermal variance term

$$\lim_{\beta \to \infty} \mathbb{E}_{\mathcal{D}} \left\langle \frac{1}{n_0} \| \mathbf{\Sigma}_0^{1/2} \mathbf{F} \delta \mathbf{v} \|^2 \right\rangle_\beta = \lim_{\beta \to \infty} \mathbb{E}_{\mathcal{D}} \frac{1}{n_0} \operatorname{tr} \left[ \mathbf{\Sigma}_0 \mathbf{F} \left( \mathbf{I}_{n_L} + \frac{\beta}{n_0} \mathbf{F}^\top \mathbf{X}^\top \mathbf{X} \mathbf{F} \right)^{-1} \mathbf{F}^\top \right]. \tag{C.12}$$

Here, we leave the thermodynamic limit implicit to allow the expression to fit on a single line.

## C.1 The overparameterized regime

First, consider the regime $p < \min\{n_0, \dots, n_L\}$. Here, we expect the kernel

$$\mathbf{K} = \frac{1}{n_0} \mathbf{X} \mathbf{F} \mathbf{F}^\top \mathbf{X}^\top \tag{C.13}$$

to be invertible with probability one in the thermodynamic limit, and with overwhelming probability at large but finite size [50]. Applying the push-through identity and passing to the ridgeless limit, we have

$$\epsilon = \lim_{\lambda \downarrow 0} \lim_{p, n_0, \dots, n_L \to \infty} \mathbb{E}_{\mathcal{D}} \frac{1}{n_0} \left\| \frac{1}{\sqrt{n_0}} \mathbf{\Sigma}_0^{1/2} \mathbf{F} \mathbf{F}^\top \mathbf{X}^\top \left( \lambda \mathbf{I}_p + \frac{1}{n_0} \mathbf{X} \mathbf{F} \mathbf{F}^\top \mathbf{X}^\top \right)^{-1} \mathbf{y} - \mathbf{\Sigma}_0^{1/2} \mathbf{w}_* \right\|^2 \tag{C.14}$$

$$= \lim_{p, n_0, \dots, n_L \to \infty} \mathbb{E}_{\mathcal{D}} \frac{1}{n_0} \left\| \sqrt{n_0} \mathbf{\Sigma}_0^{1/2} \mathbf{F} \mathbf{F}^\top \mathbf{X}^\top \left( \mathbf{X} \mathbf{F} \mathbf{F}^\top \mathbf{X}^\top \right)^{-1} \mathbf{y} - \mathbf{\Sigma}_0^{1/2} \mathbf{w}_* \right\|^2. \tag{C.15}$$

Averaging over label noise, we have

$$\epsilon = \lim_{p, n_0, \dots, n_L \to \infty} \mathbb{E}_{\mathcal{D}} \frac{1}{n_0} \left\| \mathbf{\Sigma}_0^{1/2} \mathbf{F} \mathbf{F}^\top \mathbf{X}^\top \left( \mathbf{X} \mathbf{F} \mathbf{F}^\top \mathbf{X}^\top \right)^{-1} \mathbf{X} \mathbf{w}_* - \mathbf{\Sigma}_0^{1/2} \mathbf{w}_* \right\|^2$$
$$+ \eta^2 \lim_{p, n_0, \dots, n_L \to \infty} \mathbb{E}_{\mathcal{D}} \left\| \mathbf{\Sigma}_0^{1/2} \mathbf{F} \mathbf{F}^\top \mathbf{X}^\top \left( \mathbf{X} \mathbf{F} \mathbf{F}^\top \mathbf{X}^\top \right)^{-1} \right\|^2. \tag{C.16}$$

Turning our attention to the Gibbs estimator, we can use the Woodbury identity to write the thermal variance term as

$$\mathbb{E}_{\mathcal{D}} \left\langle \frac{1}{n_0} \| \mathbf{\Sigma}_0^{1/2} \mathbf{F} \delta \mathbf{v} \|^2 \right\rangle_\beta$$

$$= \mathbb{E}_{\mathcal{D}} \frac{1}{n_0} \operatorname{tr} \left[ \mathbf{\Sigma}_0 \mathbf{F} \left( \mathbf{I}_{n_L} + \frac{\beta}{n_0} \mathbf{F}^\top \mathbf{X}^\top \mathbf{X} \mathbf{F} \right)^{-1} \mathbf{F}^\top \right] \tag{C.17}$$

$$= \mathbb{E}_{\mathcal{D}} \frac{1}{n_0} \operatorname{tr} \left[ \mathbf{\Sigma}_0 \mathbf{F} \mathbf{F}^\top \right] - \mathbb{E}_{\mathcal{D}} \frac{1}{n_0} \operatorname{tr} \left[ \mathbf{\Sigma}_0 \mathbf{F} \mathbf{F}^\top \mathbf{X}^\top \left( \beta^{-1} \mathbf{I}_{n_L} + \frac{1}{n_0} \mathbf{X} \mathbf{F} \mathbf{F}^\top \mathbf{X}^\top \right)^{-1} \mathbf{X} \mathbf{F} \mathbf{F}^\top \right] \tag{C.18}$$

$$= \mathbb{E}_{\mathcal{D}} \frac{1}{n_0} \operatorname{tr} \left[ \mathbf{\Sigma}_0 \mathbf{F} \mathbf{F}^\top \right] - \mathbb{E}_{\mathcal{D}} \frac{1}{n_0} \operatorname{tr} \left[ \mathbf{\Sigma}_0 \mathbf{F} \mathbf{F}^\top \mathbf{X}^\top \left( \frac{1}{n_0} \mathbf{X} \mathbf{F} \mathbf{F}^\top \mathbf{X}^\top \right)^{-1} \mathbf{X} \mathbf{F} \mathbf{F}^\top \right] + \mathcal{O}(\beta^{-1}), \tag{C.19}$$

where the thermodynamic limit is implied [13]. Therefore, in this regime we do not expect the thermal variance term to vanish, consistent with Proposition 6.1.

## C.2 The bottlenecked regime

If $\min\{n_1, \dots, n_L\} < \min\{n_0, p\}$, then the situation is slightly more complicated. Let

$$\ell_{\min} = \arg\min_\ell n_\ell \tag{C.20}$$

be the index of the narrowest hidden layer. Then, let

$$\mathbf{F}_1 = \frac{1}{\sqrt{n_1 \cdots n_{\ell_{\min}}}} \mathbf{U}_1 \cdots \mathbf{U}_{\ell_{\min}} \in \mathbb{R}^{n_0 \times n_{\min}} \tag{C.21}$$

and

$$\mathbf{F}_2 = \frac{1}{\sqrt{n_{\ell_{\min}+1} \cdots n_L}} \mathbf{U}_{\ell_{\min}+1} \cdots \mathbf{U}_L \in \mathbb{R}^{n_{\min} \times n_L} \tag{C.22}$$

such that

$$\mathbf{F} = \mathbf{F}_1 \mathbf{F}_2. \tag{C.23}$$

Then, the matrices $\mathbf{F}_1^\top \mathbf{X}^\top \mathbf{X} \mathbf{F}_1$ and $\mathbf{F}_2 \mathbf{F}_2^\top$ are invertible with probability one, and upon passing to the ridgeless limit we have

$$\epsilon = \lim_{p,n_0,\ldots,n_L \to \infty} \mathbb{E}_{\mathcal{D}} \frac{1}{n_0} \| \sqrt{n_0} \mathbf{\Sigma}_0^{1/2} \mathbf{F}_1 (\mathbf{F}_1^\top \mathbf{X}^\top \mathbf{X} \mathbf{F}_1)^{-1} \mathbf{F}_1^\top \mathbf{X}^\top \mathbf{y} - \mathbf{\Sigma}_0^{1/2} \mathbf{w}_* \|^2. \tag{C.24}$$

Averaging over the label noise,

$$\epsilon = \lim_{p,n_0,\ldots,n_L \to \infty} \mathbb{E}_{\mathcal{D}} \frac{1}{n_0} \| \mathbf{\Sigma}_0^{1/2} \mathbf{F}_1 (\mathbf{F}_1^\top \mathbf{X}^\top \mathbf{X} \mathbf{F}_1)^{-1} \mathbf{F}_1^\top \mathbf{X}^\top \mathbf{X} \mathbf{w}_* - \mathbf{\Sigma}_0^{1/2} \mathbf{w}_* \|^2$$
$$+ \eta^2 \lim_{p,n_0,\ldots,n_L \to \infty} \mathbb{E}_{\mathcal{D}} \| \mathbf{\Sigma}_0^{1/2} \mathbf{F}_1 (\mathbf{F}_1^\top \mathbf{X}^\top \mathbf{X} \mathbf{F}_1)^{-1} \mathbf{F}_1^\top \mathbf{X}^\top \|^2 \tag{C.25}$$

Focusing on the label noise term, we have

$$\mathbb{E}_{\mathcal{D}} \| \mathbf{\Sigma}_0^{1/2} \mathbf{F}_1 (\mathbf{F}_1^\top \mathbf{X}^\top \mathbf{X} \mathbf{F}_1)^{-1} \mathbf{F}_1^\top \mathbf{X}^\top \|^2 = \mathbb{E}_{\mathcal{D}} \operatorname{tr}[\mathbf{F}_1^\top \mathbf{\Sigma}_0 \mathbf{F}_1 (\mathbf{F}_1^\top \mathbf{X}^\top \mathbf{X} \mathbf{F}_1)^{-1}]. \tag{C.26}$$

Then, using the fact that

$$\mathbf{X}^\top \mathbf{X} \sim \mathcal{W}_{n_0}(\mathbf{\Sigma}_0, p), \tag{C.27}$$

we have

$$\mathbf{F}_1^\top \mathbf{X}^\top \mathbf{X} \mathbf{F}_1 \sim \mathcal{W}_{n_{\min}}(\mathbf{F}_1^\top \mathbf{\Sigma}_0 \mathbf{F}_1, p). \tag{C.28}$$

Then, as we expect the matrix $\mathbf{F}_1^\top \mathbf{\Sigma}_0 \mathbf{F}_1$ to be invertible with overwhelming probability, the standard formula for the mean of an inverse-Wishart distribution [50] gives

$$\mathbb{E}_{\mathcal{D}}(\mathbf{F}_1^\top \mathbf{X}^\top \mathbf{X} \mathbf{F}_1)^{-1} = \frac{1}{p - n_{\min} - 1} (\mathbf{F}_1^\top \mathbf{\Sigma}_0 \mathbf{F}_1)^{-1}, \tag{C.29}$$

so

$$\lim_{p,n_0,\ldots,n_L \to \infty} \mathbb{E}_{\mathcal{D}} \operatorname{tr}[\mathbf{F}_1^\top \mathbf{\Sigma}_0 \mathbf{F}_1 (\mathbf{F}_1^\top \mathbf{X}^\top \mathbf{X} \mathbf{F}_1)^{-1}] = \lim_{p,n_0,\ldots,n_L \to \infty} \frac{n_{\min}}{p - n_{\min} - 1} \tag{C.30}$$

$$= \frac{\alpha_{\min}}{1 - \alpha_{\min}}. \tag{C.31}$$

This proves that, in this regime, the label noise term does not depend on data structure, matching the result of our replica computation.

Considering the Gibbs estimator, we can see immediately that the thermal variance term is $\mathcal{O}(\beta^{-1})$ because of the fact that $\mathbf{F}_1^\top \mathbf{X}^\top \mathbf{X} \mathbf{F}_1$ and $\mathbf{F}_2 \mathbf{F}_2^\top$ are invertible with probability one. This is consistent with Proposition 6.1.

### C.3 The overdetermined regime

Finally, consider the regime in which $n_0 < \min\{p, n_1, \ldots, n_L\}$. Then, both $\mathbf{X}^\top \mathbf{X}$ and $\mathbf{F}\mathbf{F}^\top$ are invertible with probability one, and we can easily compute

$$\epsilon = \lim_{p,n_0,\ldots,n_L \to \infty} \lim_{\lambda \downarrow 0} \mathbb{E}_{\mathcal{D}} \frac{1}{n_0} \| \mathbf{\Sigma}_0^{1/2} (\lambda \mathbf{I}_{n_L} + \frac{1}{n_0} \mathbf{F}\mathbf{F}^\top \mathbf{X}^\top \mathbf{X})^{-1} \frac{1}{\sqrt{n_0}} \mathbf{F}\mathbf{F}^\top \mathbf{X}^\top \mathbf{y} - \mathbf{w}_* \|^2 \tag{C.32}$$

$$= \lim_{p,n_0,\ldots,n_L \to \infty} \mathbb{E}_{\mathcal{D}} \| \mathbf{\Sigma}_0^{1/2} (\mathbf{X}^\top \mathbf{X})^{-1} \mathbf{X}^\top \boldsymbol{\xi} \|^2 \tag{C.33}$$

$$= \eta^2 \lim_{p,n_0,\ldots,n_L \to \infty} \mathbb{E}_{\mathcal{D}} \operatorname{tr}[\mathbf{\Sigma}_0 (\mathbf{X}^\top \mathbf{X})^{-1}]. \tag{C.34}$$

Then,

$$(\mathbf{X}^\top \mathbf{X})^{-1} \sim \mathcal{W}_{n_0}^{-1}(\mathbf{\Sigma}_0^{-1}, p), \tag{C.35}$$

so using the formula for the mean of the inverse-Wishart [50] we have

$$\epsilon = \eta^2 \lim_{p, n_0, \dots, n_L \to \infty} \mathbb{E}_{\mathcal{D}} \operatorname{tr}[\mathbf{\Sigma}_0 (\mathbf{X}^\top \mathbf{X})^{-1}] \tag{C.36}$$

$$= \eta^2 \lim_{p, n_0, \dots, n_L \to \infty} \frac{n_0}{p - n_0 - 1} \tag{C.37}$$

$$= \frac{\alpha_0}{1 - \alpha_0} \eta^2, \tag{C.38}$$

as we found using replicas.

Here, again, we can see that the thermal variance term for the Gibbs estimator is $\mathcal{O}(\beta^{-1})$, matching Proposition 6.1.

# D  A notational dictionary

In this appendix, we show that special cases of our general result recover the results reported in previous works. This is largely a matter of translating notation, as the conventions used in different communities are often at odds with each other.

## D.1  Shallow ridgeless regression

In the shallow case $L = 0$, our general result for a fixed target (23) reduces to

$$\epsilon = \begin{cases} -\dfrac{\kappa_0^2}{\mu_0} \psi'(\kappa_0) + \dfrac{1 - \mu_0}{\mu_0} \eta^2, & \alpha_0 > 1 \\[2ex] \dfrac{\alpha_0}{1 - \alpha_0} \eta^2, & \alpha_0 < 1 \end{cases} \tag{D.1}$$

where, writing expectation with respect to the limiting spectral distribution of $\mathbf{\Sigma}_0$ as $\mathbb{E}_{\sigma_0}$, we recall that $\kappa_0$ is determined by the implicit equation

$$\frac{1}{\alpha_0} = -M_{\mathbf{\Sigma}_0}(-\kappa_0) = \mathbb{E}_{\sigma_0}\left[\frac{\sigma_0}{\kappa_0 + \sigma_0}\right], \tag{D.2}$$

in terms of which we have

$$\mu_0 = 1 - \alpha_0 \mathbb{E}_{\sigma_0}\left[\left(\frac{\sigma_0}{\kappa_0 + \sigma_0}\right)^2\right], \tag{D.3}$$

and that

$$\psi(z) = \lim_{n_0 \to \infty} \frac{1}{n_0} \mathbf{w}_*^\top \mathbf{\Sigma}_0 (z\mathbf{I}_{n_0} + \mathbf{\Sigma}_0)^{-1} \mathbf{w}_*. \tag{D.4}$$

Working in the eigenbasis of $\mathbf{\Sigma}_0$ and assuming that $\|\mathbf{w}_*\|^2 = n_0$, we introduce the weighted density

$$\rho(\sigma_0) = \lim_{n_0 \to \infty} \frac{1}{n_0} \sum_{j=1}^{n_0} (w_*)_j^2 \delta(\sigma_0 - \sigma_j) \tag{D.5}$$

in terms of which we have

$$\psi(z) = \mathbb{E}_{\sigma_0 \sim \rho}\left[\frac{\sigma_0}{z + \sigma_0}\right] \tag{D.6}$$

and

$$-\psi'(z) = \mathbb{E}_{\sigma_0 \sim \rho}\left[\frac{\sigma_0}{(z + \sigma_0)^2}\right]. \tag{D.7}$$

We can now make contact with the result of Hastie et al. [5]. We note that those authors use an opposite definition for $p$ and $n$: following the convention in the statistics literature, they use $p$ for

the dimensionality and $n$ for the number of examples, while we follow the convention in the physics literature of using $n_0$ for the dimensionality and $p$ for the number of examples. Then, Hastie et al. [5]'s $\gamma$, defined such that, in our terms, $n_0/p \to \gamma$, is precisely our $\alpha_0$. Moreover, they use $H(z)$ to denote the limiting spectral law of $\Sigma_0$, and $G(z)$ to denote the law corresponding to the weighted density we define above as $\rho$. We note also that their $\sigma^2$ is our $\eta^2$. In these terms, their Theorem 2 gives the generalization error in the overparameterized regime $\alpha_0 > 1$ as

$$\epsilon = \left\{ 1 + \alpha_0 c_0 \frac{\mathbb{E}_{\sigma_0}\left[\frac{\sigma_0^2}{(1+c_0\alpha_0\sigma_0)^2}\right]}{\mathbb{E}_{\sigma_0}\left[\frac{\sigma_0}{(1+c_0\alpha_0\sigma_0)^2}\right]} \right\} \mathbb{E}_{\sigma_0 \sim \rho}\left[\frac{\sigma_0}{(1+c_0\alpha_0\sigma_0)^2}\right] + \eta^2 \alpha_0 c_0 \frac{\mathbb{E}_{\sigma_0}\left[\frac{\sigma_0^2}{(1+c_0\alpha_0\sigma_0)^2}\right]}{\mathbb{E}_{\sigma_0}\left[\frac{\sigma_0}{(1+c_0\alpha_0\sigma_0)^2}\right]} \quad \text{(D.8)}$$

where $c_0$ is defined by the implicit equation

$$1 - \frac{1}{\alpha_0} = \mathbb{E}_{\sigma_0}\left[\frac{1}{1+c_0\alpha_0\sigma_0}\right]. \quad \text{(D.9)}$$

Subtracting one from both sides, the implicit equation for $c_0$ gives

$$\frac{1}{\alpha_0} = \mathbb{E}_{\sigma_0}\left[\frac{c_0\alpha_0\sigma_0}{1+c_0\alpha_0\sigma_0}\right] \quad \text{(D.10)}$$

from which we can see that

$$c_0\alpha_0 = \frac{1}{\kappa_0}. \quad \text{(D.11)}$$

Then, we have

$$\mathbb{E}_{\sigma_0 \sim \rho}\left[\frac{\sigma_0}{(1+c_0\alpha_0\sigma_0)^2}\right] = \kappa_0^2 \mathbb{E}_{\sigma_0 \sim \rho}\left[\frac{\sigma_0}{(\kappa_0+\sigma_0)^2}\right] \quad \text{(D.12)}$$

$$= -\kappa_0^2 \psi'(\kappa_0) \quad \text{(D.13)}$$

and

$$\alpha_0 c_0 \frac{\mathbb{E}_{\sigma_0}\left[\frac{\sigma_0^2}{(1+c_0\alpha_0\sigma_0)^2}\right]}{\mathbb{E}_{\sigma_0}\left[\frac{\sigma_0}{(1+c_0\alpha_0\sigma_0)^2}\right]} = \frac{\mathbb{E}_{\sigma_0}\left[\frac{(\alpha_0 c_0\sigma_0)^2}{(1+c_0\alpha_0\sigma_0)^2}\right]}{\mathbb{E}_{\sigma_0}\left[\frac{\alpha_0 c_0\sigma_0}{(1+c_0\alpha_0\sigma_0)^2}\right]} \quad \text{(D.14)}$$

$$= \frac{\mathbb{E}_{\sigma_0}\left[\frac{\sigma_0^2}{(\kappa_0+\sigma_0)^2}\right]}{\mathbb{E}_{\sigma_0}\left[\frac{\kappa_0\sigma_0}{(\kappa_0+\sigma_0)^2}\right]} \quad \text{(D.15)}$$

$$= \frac{1-\mu_0}{\alpha_0 \mathbb{E}_{\sigma_0}\left[\frac{\kappa_0\sigma_0}{(\kappa_0+\sigma_0)^2}\right]} \quad \text{(D.16)}$$

$$= \frac{1-\mu_0}{\alpha_0 \mathbb{E}_{\sigma_0}\left[\frac{\sigma_0}{\kappa_0+\sigma_0}\right] - \alpha_0 \mathbb{E}_{\sigma_0}\left[\frac{\sigma_0^2}{(\kappa_0+\sigma_0)^2}\right]} \quad \text{(D.17)}$$

$$= \frac{1-\mu_0}{\mu_0}, \quad \text{(D.18)}$$

which proves the equivalence of our results. This also shows that we recover the results of other works on ridgeless kernel interpolation [4, 6, 7, 16, 17] that are in this setting equivalent to the results of Hastie et al. [5].

## D.2 Two-layer linear random feature models with unstructured weights and isotropic targets

Another special case in which we can make contact with prior work is that of a single hidden layer ($L = 1$) and with target averaging. In this case, our general result (27) reduces to

$$\bar{\epsilon} = \begin{cases} \left(1 + \frac{1}{\alpha_1 - 1}\right)\chi(\alpha_0) + \left(\frac{1-\mu_0}{\mu_0} + \frac{1}{\alpha_0 - 1}\right)\eta^2 & \alpha_0, \alpha_1 > 1 \\ \frac{1}{1-\alpha_1}\chi\left(\frac{\alpha_0}{\alpha_1}\right) + \frac{\alpha_1}{1-\alpha_1}\eta^2 & \alpha_1 < 1, \alpha_1 < \alpha_0 \\ \frac{\alpha_0}{1-\alpha_0}\eta^2 & \alpha_0 < 1, \alpha_0 < \alpha_1 \end{cases} \quad \text{(D.19)}$$

where in this case we find it convenient to write $\kappa_0/\alpha_0$ and $\alpha_0\kappa_{\min}/\alpha_{\min}$ in terms of $\chi(z)$, which solves

$$1 = -z M_{\tilde{\boldsymbol{\Sigma}}_0}[-z\chi(z)] \tag{D.20}$$

$$= \mathbb{E}_{\tilde{\sigma}_0}\left[\frac{\tilde{\sigma}_0}{\chi(z) + z^{-1}\tilde{\sigma}_0}\right]. \tag{D.21}$$

It is then easy to show that our result agrees with that of Maloney et al. [19]. Their notation is:

$$M = n_0 \tag{D.22}$$
$$N = n_1 \tag{D.23}$$
$$T = p. \tag{D.24}$$

When $M > N, T$, their result is, in the absence of label noise,

$$\bar{\epsilon} = \frac{1}{M}\begin{cases}\dfrac{1}{1 - N/T}\Delta_{-1}(N, M), & N < T \\[2ex] \dfrac{1}{1 - T/N}\Delta_{-1}(T, M), & N > T,\end{cases} \tag{D.25}$$

where $\Delta_{-1}(N, M)$ solves

$$1 = \mathrm{tr}[\boldsymbol{\Sigma}_0(\Delta_{-1}(N, M)\mathbf{I}_M + N\boldsymbol{\Sigma}_0)^{-1}] \tag{D.26}$$

and similarly for $\Delta_{-1}(T, M)$. To map this to our results, let us re-define

$$\bar{\Delta}_{-1}(N, M) \equiv \frac{1}{M}\Delta_{-1}(N, M), \tag{D.27}$$

which then satisfies

$$1 = \frac{1}{M}\mathrm{tr}[\boldsymbol{\Sigma}_0(\bar{\Delta}_{-1}(N, M)\mathbf{I}_M + (N/M)\boldsymbol{\Sigma}_0)^{-1}], \tag{D.28}$$

or

$$\frac{N}{M} = -M_{\boldsymbol{\Sigma}_0}\left(-\frac{M}{N}\bar{\Delta}_{-1}(N, M)\right) \tag{D.29}$$

Then, we can see that, in our notation,

$$\bar{\Delta}_{-1}(N, M) = \chi\left(\frac{M}{N}\right) = \chi\left(\frac{\alpha_0}{\alpha_1}\right), \tag{D.30}$$

while

$$\bar{\Delta}_{-1}(T, M) = \chi\left(\frac{M}{T}\right) = \chi(\alpha_0). \tag{D.31}$$

Then, noting that

$$\frac{1}{1 - T/N} = \frac{1}{1 - 1/\alpha_1} = \frac{\alpha_1}{\alpha_1 - 1} = 1 + \frac{1}{\alpha_1 - 1}, \tag{D.32}$$

we can see that we recover their result in these regimes. We can also map their $\Delta_0$ to our $\mu_0$. For $T < M$, they let

$$\frac{\Delta_0}{1 + \Delta_0} = \sum_{j=1}^{M}\frac{T\sigma_j^2}{(T\sigma_j + \Delta_{-1})^2} \tag{D.33}$$

$$= \frac{1}{T}\sum_{j=1}^{M}\frac{\sigma_j^2}{(\sigma_j + M/T\bar{\Delta}_{-1})^2}, \tag{D.34}$$

hence we can see that

$$\frac{\Delta_0}{1 + \Delta_0} = 1 - \mu_0. \tag{D.35}$$

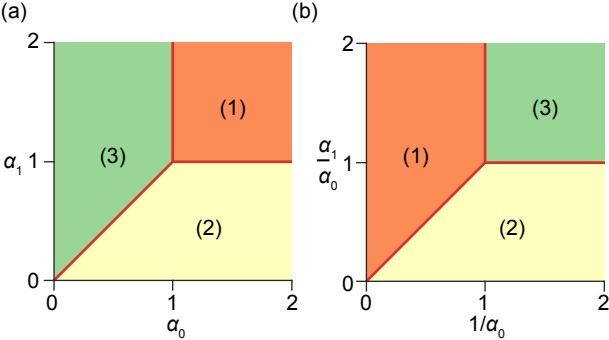

Figure D.1: Phase diagrams in different parameterizations of the thermodynamic limit. (a). Phase diagram in the $(\alpha_0, \alpha_1)$ plane. Region 1 (*orange*) is the overparameterized regime, Region 2 (*yellow*) is the bottlenecked regime, and Region 3 (*green*) is the overdetermined regime. (b). As in (a), but in the $(1/\alpha_0, \alpha_1/\alpha_0)$ plane, matching the parameterization used in our previous work [13]. Note that the plane is divided identically, but the locations of the phases are swapped.

This mapping also enables our application of their interpolating approximate solutions for $\Delta_{-1}$ and $\Delta_0$ in the case of power law spectra. For a finite-size spectrum

$$\sigma_j = \frac{\sigma_+}{j^{1+\omega}} \qquad (j = 1, \ldots, M), \tag{D.36}$$

with

$$\sigma_+ = M^{1+\omega}\sigma_-, \tag{D.37}$$

where we denote the exponent by $\omega$ rather than $\alpha$ as Maloney et al. [19] do to avoid clashing with our notation elsewhere, they obtain the approximate solution

$$\frac{1}{M}\Delta_{-1}(N, M) = \begin{cases} \sigma_- \left\{ k \left[ \left( \frac{M}{N} \right)^\omega - 1 \right] + [2 + \omega(1 - k)] \left( 1 - \frac{N}{M} \right) \right\}, & N < M \\ 0 & N > M \end{cases} \tag{D.38}$$

for

$$k = \left[ \frac{\frac{\pi}{1+\omega}}{\sin\left(\frac{\pi}{1+\omega}\right)} \right]^{1+\omega} = \left[ \frac{1}{\mathrm{sinc}\left(\frac{\pi}{1+\omega}\right)} \right]^{1+\omega}, \tag{D.39}$$

which leads to the expression

$$\chi(z) = \begin{cases} \sigma_- \left\{ k(z^\omega - 1) + [2 + \omega(1 - k)] \left( 1 - \frac{1}{z} \right) \right\}, & z > 1 \\ 0 & z < 1. \end{cases} \tag{D.40}$$

Moreover, for $T < M$, they give the approximate solution

$$\Delta_0(T, M) = \omega + \frac{1}{M/T - 1}. \tag{D.41}$$

By applying these results, we obtain the result claimed in the main text, (31). We note that we fix $\sigma_-$ to be constant rather than $\sigma_+$ as Maloney et al. [19] do, which ensures normalizability of the limiting eigenvalue distribution at the expense of diverging moments.

### D.3 Deep linear models with unstructured weights and data

In [13], we studied deep Bayesian linear models with unstructured features and data.[2] There, and in very recent work by Schröder et al. [20], a different parameterization for the thermodynamic limit

---

[2]In [13], we focused on the Gibbs estimator rather than on the ridgeless maximum-likelihood estimator (MLE). However, given the average generalization error for the Gibbs estimator, it is easy to obtain the generalization error for the MLE. We discuss this point in detail in Appendix A.

was used:

$$p, n_0, \ldots, n_L \to \infty, \quad \text{with} \quad \frac{p}{n_0} \to \tilde{\alpha}, \quad \frac{n_\ell}{n_0} \to \tilde{\gamma}_\ell \quad (\ell = 1, \ldots, L), \tag{D.42}$$

where we decorate $\tilde{\alpha}$ and $\tilde{\gamma}_\ell$ with tildes to avoid confusion with parameters used elsewhere in the present work. The conversion to the parameterization used in the present work and in [30] is then given by

$$\tilde{\alpha} = \frac{1}{\alpha_0}, \tag{D.43}$$

$$\tilde{\gamma}_\ell = \frac{\alpha_\ell}{\alpha_0}, \qquad (\ell = 1, \ldots, L). \tag{D.44}$$

Though these parameterizations are mathematically equivalent, it is important to distinguish between them as they give phase diagrams that divide the plane identically but swap the locations of the phases, as is shown in Figure D.1. Moreover, though the parameterization used here is more convenient for the replica computation [30], that given in (D.42) is conceptually useful, as it is closer to what one does in practical machine learning settings: the input dimension $n_0$ is fixed by the task, and one can vary the dataset size $p$ and the network widths $n_\ell$. This is why we plot the phase diagrams in Figure 1 in the $(1/\alpha_0, \alpha_1/\alpha_0)$ plane.

# E   Large-width expansions

In this appendix, we consider the limit of large width, i.e., the limit in which $\alpha_1, \ldots, \alpha_L \to \infty$ for fixed $\alpha_0$. Our first task is to determine how the quantities $\kappa_\ell$ behave in this limit, as it is through these inverse generating functions that the hidden layer widths enter the generalization error.

Starting from the defining equation

$$\frac{1}{\alpha_\ell} = \mathbb{E}_{\tilde{\sigma}_\ell} \left[ \frac{\tilde{\sigma}_\ell}{\kappa_\ell + \tilde{\sigma}_\ell} \right] \tag{E.1}$$

we can see that $\kappa_\ell$ should tend to infinity linearly with $\alpha_\ell$ as $\alpha_\ell \to \infty$. In particular, we should have

$$\frac{\kappa_\ell}{\alpha_\ell} \to \mathbb{E}_{\tilde{\sigma}_\ell}[\tilde{\sigma}_\ell] \tag{E.2}$$

at large widths. Then, $\mu_\ell$ has limiting behavior

$$\mu_\ell = 1 - \alpha_\ell \mathbb{E}_{\tilde{\sigma}_\ell} \left[ \left( \frac{\tilde{\sigma}_\ell}{\kappa_\ell + \tilde{\sigma}_\ell} \right)^2 \right] \tag{E.3}$$

$$\to 1 \tag{E.4}$$

From this, we can see that in the infinite-width limit the generalization error of the random feature model in Proposition 3.2 reduces to that of shallow ridgeless regression as in Corollary 3.1, as we would expect.

We now want to compute the leading correction to this result. In the unstructured case, this is easy, because we have $\kappa_\ell = (\alpha_\ell - 1)\tilde{\sigma}_\ell$, hence there is an $\mathcal{O}(1)$ correction and nothing else. More generally, we assume Laurent series behavior of the form

$$\kappa_\ell = \alpha_\ell \kappa_\ell^1 + \kappa_\ell^0 + \frac{1}{\alpha_\ell} \kappa_\ell^{-1} + \ldots. \tag{E.5}$$

Expanding, we have

$$\frac{\tilde{\sigma}_\ell}{\kappa_\ell + \tilde{\sigma}_\ell} = \frac{\tilde{\sigma}_\ell}{\alpha_\ell \kappa_\ell^1} - \frac{\tilde{\sigma}_\ell(\tilde{\sigma}_\ell + \kappa_\ell^0)}{\alpha_\ell^2 (\kappa_\ell^1)^2} + \mathcal{O}(\alpha_\ell^{-3}) \tag{E.6}$$

hence, if we integrate term-by-term, we have

$$\frac{1}{\alpha_\ell} = \frac{\mathbb{E}_{\tilde{\sigma}_\ell}[\tilde{\sigma}_\ell]}{\alpha_\ell \kappa_\ell^1} - \frac{\mathbb{E}_{\tilde{\sigma}_\ell}[\tilde{\sigma}_\ell^2] + \mathbb{E}_{\tilde{\sigma}_\ell}[\tilde{\sigma}_\ell]\kappa_\ell^0}{\alpha_\ell^2 (\kappa_\ell^1)^2} + \mathcal{O}(\alpha_\ell^{-3}). \tag{E.7}$$

If we solve order-by-order, we again find that

$$\kappa_\ell^1 = \mathbb{E}_{\tilde\sigma_\ell}[\tilde\sigma_\ell] \tag{E.8}$$

while the coefficients of all higher-order terms in $1/\alpha_\ell$ must vanish. In particular, this gives

$$\kappa_\ell^0 = -\frac{\mathbb{E}_{\tilde\sigma_\ell}[\tilde\sigma_\ell^2]}{\mathbb{E}_{\tilde\sigma_\ell}[\tilde\sigma_\ell]}. \tag{E.9}$$

This computation assumes that the spectrum has finite moments, which is not the case for the heavy-tailed power law spectra considered in Corollary 5.1.

Then, we have

$$\mu_\ell = 1 - \alpha_\ell \mathbb{E}_{\tilde\sigma_\ell}\left[\left(\frac{\tilde\sigma_\ell}{\kappa_\ell + \tilde\sigma_\ell}\right)^2\right] \tag{E.10}$$

$$= 1 - \frac{\mathbb{E}_{\tilde\sigma_\ell}[\tilde\sigma_\ell^2]}{\alpha_\ell(\kappa_\ell^1)^2} + 2\frac{\mathbb{E}_{\tilde\sigma_\ell}[\tilde\sigma_\ell^3] + \mathbb{E}_{\tilde\sigma_\ell}[\tilde\sigma_\ell^2]\kappa_\ell^0}{\alpha_\ell^2(\kappa_\ell^1)^3} + \mathcal{O}(\alpha_\ell^{-3}) \tag{E.11}$$

$$= 1 - \frac{\mathbb{E}_{\tilde\sigma_\ell}[\tilde\sigma_\ell^2]}{\mathbb{E}_{\tilde\sigma_\ell}[\tilde\sigma_\ell]^2}\frac{1}{\alpha_\ell} + \mathcal{O}(\alpha_\ell^{-2}). \tag{E.12}$$

Collecting our results, we have

$$\kappa_\ell = \mathbb{E}_{\tilde\sigma_\ell}[\tilde\sigma_\ell]\alpha_\ell\left(1 - \frac{\mathbb{E}_{\tilde\sigma_\ell}[\tilde\sigma_\ell^2]}{\mathbb{E}_{\tilde\sigma_\ell}[\tilde\sigma_\ell]^2}\frac{1}{\alpha_\ell} + \mathcal{O}(\alpha_\ell^{-2})\right) \tag{E.13}$$

and

$$\mu_\ell = 1 - \frac{\mathbb{E}_{\tilde\sigma_\ell}[\tilde\sigma_\ell^2]}{\mathbb{E}_{\tilde\sigma_\ell}[\tilde\sigma_\ell]^2}\frac{1}{\alpha_\ell} + \mathcal{O}(\alpha_\ell^{-2}). \tag{E.14}$$

Each term in these expansions has the expected behavior under rescaling: if we let $\tilde\Sigma_\ell' = \tau_\ell\tilde\Sigma_\ell$ for $\tau_\ell > 0$, we have $\kappa_\ell' = \tau_\ell\kappa_\ell$ and $\mu_\ell' = \mu_\ell$.

Then, substituting these expansions into (23), we find that the generalization error of an RFM in the ridgeless limit expands at large widths as

$$\epsilon = -\frac{\kappa_0^2}{\mu_0}\psi'(\kappa_0) + \frac{1 - \mu_0}{\mu_0}\eta^2$$
$$+ \left(\sum_{\ell=1}^L \frac{\mathbb{E}_{\tilde\sigma_\ell}[\tilde\sigma_\ell^2]}{\mathbb{E}_{\tilde\sigma_\ell}[\tilde\sigma_\ell]^2}\frac{1}{\alpha_\ell}\right)(\kappa_0\psi(\kappa_0) + \eta^2)$$
$$+ \mathcal{O}(\alpha_1^{-2}, \ldots, \alpha_L^{-2}) \tag{E.15}$$

in the regime $\alpha_0 > 1$; if $\alpha_0 < 1$ the generalization error does not depend on the hidden layer widths so long as they are greater than 1.

For an RFM trained using the Gibbs estimator, as considered in Proposition 6.1, we find that

$$\epsilon_{\mathrm{BRFM}} = -\frac{\kappa_0^2}{\mu_0}\psi'(\kappa_0) + \frac{1 - \mu_0}{\mu_0}\eta^2 + \frac{\kappa_0}{\alpha_0}\varsigma^2$$
$$+ \left(\sum_{\ell=1}^L \frac{\mathbb{E}_{\tilde\sigma_\ell}[\tilde\sigma_\ell^2]}{\mathbb{E}_{\tilde\sigma_\ell}[\tilde\sigma_\ell]^2}\frac{1}{\alpha_\ell}\right)\left(\kappa_0\psi(\kappa_0) + \eta^2 - \frac{\kappa_0}{\alpha_0}\varsigma^2\right)$$
$$+ \mathcal{O}(w^{-2}) \tag{E.16}$$

where we have defined

$$\varsigma^2 \equiv \prod_{\ell=1}^L \mathbb{E}_{\tilde\sigma_\ell}[\tilde\sigma_\ell], \tag{E.17}$$

upon expanding the thermal variance term

$$
\prod_{\ell=0}^{L} \frac{\kappa_\ell}{\alpha_\ell} = \frac{\kappa_0}{\alpha_0} \left[ \prod_{\ell=1}^{L} \mathbb{E}_{\tilde{\sigma}_\ell}[\tilde{\sigma}_\ell] \right]
$$

$$
- \frac{\kappa_0}{\alpha_0} \left[ \prod_{\ell=1}^{L} \mathbb{E}_{\tilde{\sigma}_\ell}[\tilde{\sigma}_\ell] \right] \sum_{\ell=1}^{L} \frac{\mathbb{E}_{\tilde{\sigma}_\ell}[\tilde{\sigma}_\ell^2]}{\mathbb{E}_{\tilde{\sigma}_\ell}[\tilde{\sigma}_\ell]^2} \frac{1}{\alpha_\ell}
$$

$$
+ \mathcal{O}(w^{-2}). \tag{E.18}
$$

Here, we denote by $\mathcal{O}(w^{-2})$ all terms of $\mathcal{O}(\alpha_\ell^{-2})$ for a given layer $\ell = 1, \ldots, L$ or terms of $\mathcal{O}(\alpha_\ell^{-1}\alpha_{\ell'}^{-1})$ for two different layers $\ell, \ell'$.

# F  Numerical methods

In this appendix, we describe the numerical methods used to produce Figures 1, 2. All simulations were performed using MATLAB 9.13 (R2022b; The MathWorks, Natick MA, USA; https://www.mathworks.com/products/matlab.html) on a desktop workstation (CPU: Intel Xeon W-2145, 64GB RAM). They were not computationally intensive, and required less than an hour of compute time in total. Code to reproduce the figures is archived as part of the online supplemental material. Numerical computation of the solution to the ridgeless regression problem—the minimum-norm interpolant—was performed using the lsqminnorm solver (https://www.mathworks.com/help/matlab/ref/lsqminnorm.html), which uses an algorithm based on the complete orthogonal decomposition of the design matrix.

