# OpenReview forum: "Learning Curves for Deep Structured Gaussian Feature Models"
_NeurIPS.cc/2023/Conference — NeurIPS 2023 poster_

### Official Review · Reviewer_VEZW · 2023-07-04

**Soundness:** 4 excellent
**Presentation:** 3 good
**Contribution:** 3 good
**Rating:** 7
**Confidence:** 3

**Summary:**

This work focuses on the generalization performance of models utilizing multi-layered Gaussian random features. The study evaluates the impact of feature anisotropy, which is often overlooked due to common assumptions that features are generated using independent, identically distributed Gaussian weights. The findings demonstrate that correlations within the first layer of features can enhance generalization, but any structure beyond the initial layer proves generally detrimental. These insights provide valuable perspective on how weight structure affects generalization in random feature models with linear activations.


**Strengths:**

1. The paper is well-written and the theorems are constructed with a solid use of mathematical rigor.

2. The idea is interesting, where how correlation between the rows of the first layer of features can improve generalization is a new theoretical result, which could be very interesting to the community.

3. Visualization and numerical experiments are conducted along with the theoretical results. For instance, the experiments in Figure 2 nicely summarize the theoretical finding, implying the bounds are generally non-vacuous.

**Weaknesses:**

1. The discussion to previous works are limited. There are many RFM works recently. Although they may focus on a different perspective, such as inductive biases or behavior under SGD, but it would still be valuable to briefly discuss the relationship to these works.

2. At the first glimpse, it is hard to parse how the structure can improve the generalization in some of the theorem or corollary. In particular, there are many constants and it is non-trivial to interpret their numerical property. Adding a simple sentence after the theorem to briefly summarize it could help reader better parse the results.


**Questions:**

1. Would figure 2 vary a lot if we modified the number of data points?  In particular, would the theory behaves differently under over-parameterized or under-parameterized regime?

2. The theorems analyze the asymptotic learning curve, where the numerical experiments are conducted under a relatively small number of datapoints. Will this cause a gap between the theory and the experimental results?

3. The constraint on the norm of the teacher vector (line 142) is not very intuitive. Can you briefly explain this?


4. Does the result approximately hold for setting without closed form such as Lasso-regression?



**Limitations:**

The authors did discuss the limitation and there is no obvious negative societal impact.

---

> ### Author Rebuttal · Authors · 2023-08-05
>
> We thank the reviewer for their careful assessment of our work, and are gratified by their favorable comments.
>
> ### Weaknesses:
>
> 1. *The discussion to previous works are limited. There are many RFM works recently. Although they may focus on a different perspective, such as inductive biases or behavior under SGD, but it would still be valuable to briefly discuss the relationship to these works.*
>
> Thank you for this suggestion. As mentioned in our common response to referees, we will add a more extensive discussion of related works on RFMs to the Introduction.
>
> 2. *At the first glimpse, it is hard to parse how the structure can improve the generalization in some of the theorem or corollary. In particular, there are many constants and it is non-trivial to interpret their numerical property. Adding a simple sentence after the theorem to briefly summarize it could help reader better parse the results.*
>
> Thank you for this helpful suggestion. We will defer most proofs to the Appendix, and will add expository sentences to summarize each.
>
> ### Questions:
>
> 1. *Would figure 2 vary a lot if we modified the number of data points? In particular, would the theory behaves differently under over-parameterized or under-parameterized regime?*
>
> Thank you for this question. Figure 2 shows precisely a sweep over the number of datapoints: $1/\alpha_{0}$ is the ratio of the number of datapoints to the input dimensionality. Therefore, as the abscissa of each of the panels of Figure 2 increases, the number of datapoints increases. These figures, and also the general form of our results, include the transition from over-parameterized to under-parameterized regimes.
>
> 2. *The theorems analyze the asymptotic learning curve, where the numerical experiments are conducted under a relatively small number of datapoints. Will this cause a gap between the theory and the experimental results?*
>
> Finite-size experiments should certainly deviate by some amount from the asymptotic approximation. As illustrated by Figure 2, the rate of convergence is sufficiently fast such that the agreement is good even at these sizes.
>
> 3. *The constraint on the norm of the teacher vector (line 142) is not very intuitive. Can you briefly explain this?*
>
> The constraint on the norm of the teacher weight vector is a matter of convenience, and amounts to a choice of units for the label noise and generalization error. The important assumption is that $\Vert \tilde{\mathbf{w}}\_{\ast} \Vert^2 / n\_{0}$ tends to a constant as $n_{0} \to \infty$. If this constant is $C$ rather than 1, the generalization error for isotropic spectra (as in Corollary 3.3) would be the same as that for $C = 1$ if we re-normalize the noise strength as $\eta^2/C$ and the generalization error as $\epsilon/C$. Thus, this convention is a matter of convenience.
>
> 4. *Does the result approximately hold for setting without closed form such as Lasso-regression?*
>
> Thank you for this question. We have not tested other settings, but we certainly agree that extending these results to other convex regularizers would be an interesting topic for future work.

---

> > ### Comment · Reviewer_VEZW · 2023-08-19
> > **Response**
> >
> > Thank you for your explanation which partially addresses the raised concerns. Overall, I think the proposed ideas are interesting, and I will keep my score.

---

### Official Review · Reviewer_nhrG · 2023-07-06

**Soundness:** 3 good
**Presentation:** 1 poor
**Contribution:** 3 good
**Rating:** 5
**Confidence:** 3

**Summary:**

The authors investigate the exact asymptotics characterization of deep Gaussian feature models. They analyze depth-L linear Random Feature Models where the feature matrix is built as a product of L factors, each drawn from a matrix Gaussian distribution. Thanks to the replica method, they compute the generalization error learning curves for the ridge regression estimator. They study the influence of the weight structure on the test performance, presenting numerical experiments backing up the theoretical claims.

**Strengths:**

The authors consider the interesting case of deep RFMs with weight anisotropy. They compute the performance for both ERM and Bayes estimators in the high-dimensional proportional regime. They present numerical simulation and release the code to reproduce the main experiments.

**Weaknesses:**

The primary weakness of this paper is the clarity of the presentation, more precisely I believe more effort should be put into guiding a non-expert reader with a more detailed introduction, and by creating connections between the different main results presented. See below for a more detailed discussion.

**Questions:**

1) The abstract is very concise. I would expand this section to help guide the reader, e.g., mention that you focus on interpolators,  describe what exactly solvable model for anisotropic spectra you considered (power law), briefly explain that the proposed formalism allows as well to cover analysis of Bayesian setting.
2) On a similar note to the first point, the introduction section is too short. There is little explanation of why RFMs are interesting, e.g., it is not mentioned explicitly their relation with the limiting kernel. The analysis of power-law spectrum decay is standard in the kernel literature and goes under the name of source-capacity conditions [1]. There is no mention of the technique which is going to be used for the computation (the replica trick). Please mention why the interpolating (ridgeless) regime is interesting, with associated reference, e.g. ref [5] in the main text. There is no citation to previous work on the Bayesian settings, the reference [42] mentioned in the conclusion could be introduced as well there. A justification should be given for the Gaussian data assumption, e.g., by referring to the extensive line of work on the Gaussian Universality property, see [2].
3) Please before introducing the preliminaries and the setting on page 2, include a summary of your main contributions.
4) Why is it untractable to study the $\lambda > 0$ case? A numerical investigation of the behavior at finite regularization I believe would enhance the manuscript, if the problem is not untractable. If this is the case, please mention why in the main text.
5) The results are nice but I believe some work is needed to glue them together. The authors may think to move in the appendix some proof and substitute them with explanations in plain words to help the reader build intuition, as the notation is quite heavy.
6) In Sec. 4 remind the reader in which theorem are defined $(\alpha_{min},\mu)$.
7) Can the authors explain in more detail why the exponents $(\omega_1,\dots,\omega_L)$ do not affect the scaling laws if they enter only with the sum (Sec. 5)?

[1] Andrea Caponnetto and Ernesto De Vito. Optimal rates for the regularized least-squares algorithm, 2007.

[2] Montanari, A. and Saeed, B. N. Universality of empirical risk minimization, 2022.

**Limitations:**

The limitations are addressed in the manuscript.

---

> ### Author Rebuttal · Authors · 2023-08-05
>
> We sincerely appreciate the referee’s feedback on the clarity of our submitted manuscript, and will revise our manuscript in accordance with their valuable suggestions.
>
> ### Questions:
> 1. *The abstract is very concise. I would expand this section to help guide the reader, e.g., mention that you focus on interpolators, describe what exactly solvable model for anisotropic spectra you considered (power law), briefly explain that the proposed formalism allows as well to cover analysis of Bayesian setting. *
>
> As mentioned in our common response, we will revise the Abstract following your suggestions.
>
> 2. *On a similar note to the first point, the introduction section is too short. There is little explanation of why RFMs are interesting, e.g., it is not mentioned explicitly their relation with the limiting kernel. The analysis of power-law spectrum decay is standard in the kernel literature and goes under the name of source-capacity conditions [1]. There is no mention of the technique which is going to be used for the computation (the replica trick). Please mention why the interpolating (ridgeless) regime is interesting, with associated reference, e.g. ref [5] in the main text. There is no citation to previous work on the Bayesian settings, the reference [42] mentioned in the conclusion could be introduced as well there. A justification should be given for the Gaussian data assumption, e.g., by referring to the extensive line of work on the Gaussian Universality property, see [2].*
>
> Thank you for these suggestions. As mentioned in our common response, we will expand the Introduction to address each of these points in detail. We particularly appreciate the referee’s reference suggestions, which we unfortunately missed in the submitted manuscript.
>
> 3. *Please before introducing the preliminaries and the setting on page 2, include a summary of your main contributions.*
>
> As mentioned in our common response, we will add a bulleted list of our primary contributions. In short, this list is:
>
> - Using the replica method from statistical mechanics, we compute the asymptotic generalization error of deep linear random feature models with weights drawn from general matrix Gaussian distributions.
> - We show that, in the ridgeless limit, structure in the weights beyond the first layer is detrimental for generalization.
> - Focusing on the approximately solvable special case of power-law spectra in the weights and in the data, we show that the weight spectrum power laws do not affect the scaling laws of generalization.
> - We show how our results can be extended from the ridge regression estimator to the Bayesian Gibbs estimator. For sufficiently large prior variance, structure can be beneficial for generalization with this estimator.
>
> 4. *Why is it untractable to study the $\lambda > 0$ case? A numerical investigation of the behavior at finite regularization I believe would enhance the manuscript, if the problem is not untractable. If this is the case, please mention why in the main text.*
>
> We wholeheartedly agree that a careful investigation of the $\lambda > 0$ case would be interesting. Please see our common response regarding this concern. We reiterate from there that to the best of our knowledge a detailed numerical investigation at large depths ($L>2$) would be challenging, based on analogy with work on product random matrices.
>
> 5. *The results are nice but I believe some work is needed to glue them together. The authors may think to move in the appendix some proof and substitute them with explanations in plain words to help the reader build intuition, as the notation is quite heavy.*
>
> Thank you for this suggestion. We will move the proofs to the appendix, and use the space thusly freed to add expanded discussions of each result. Please see our common response to comments for a detailed plan of revisions.
>
> 6. *In Sec. 4 remind the reader in which theorem are defined ($\alpha_{\mathrm{min}}, \mu$).*
>
> Thank you, we will fix this.
>
> 7. *Can the authors explain in more detail why the exponents $(\omega_{1}, \ldots, \omega_{L})$ do not affect the scaling laws if they enter only with the sum (Sec. 5)?*
>
> Thank you for this question. Our reasoning here is that the weight exponents do not affect the scaling when $\alpha_{0} \gg 1$, as in equation (32). Here, we are using the convention from Maloney et al. (2022) that the scaling law is determined asymptotically. Therefore, we discard all sub-leading contributions and also neglect the exponent-dependent constant multiplying $\alpha_{0}^{\omega_{0}}$.

---

> > ### Comment · Reviewer_nhrG · 2023-08-20
> > **Thank you for the rebuttal**
> >
> > I sincerely thank the authors for the rebuttal.
> >
> >
> > I think that the main weakness is the clarity in the presentation and that the promised changes will strongly help the readability of the text. I would like to keep my score as in the original review.

---

### Official Review · Reviewer_cGDX · 2023-07-13

**Soundness:** 3 good
**Presentation:** 2 fair
**Contribution:** 3 good
**Rating:** 6
**Confidence:** 1

**Summary:**

The authors introduce correlations to weights of linear RFMs and study the generalization error.

This is done under the assumption data follows Gaussian distribution and linear model.

The authors provide a general expression for the limiting generalization error that recovers results from previous works as special cases.

**Strengths:**

The present results present appear to be very general and well linked to previous research.

The introduction and preliminaries section are relatively clear.

**Weaknesses:**

The proofs could be included in the supplementary material which would allow for more text interpreting the results and providing the intuition for their importance.

The authors should explain the solutions (15) and (16).

The authors should explain in more detail their result, i.e. (18).

The authors should be more explicit about the notation, e.g. carefully check the use of /tilde or $\kappa_0$.

In my opinion, the clarity of the paper could be improved.

**Questions:**

Do solutions (15) and (16) always exist? Can you elaborate on their form?

Do you have any intuition how quickly convergence to (13) happen?

Do you have any intuition about the limits when p is fixed?

---

> ### Author Rebuttal · Authors · 2023-08-05
>
> Thank you for your careful assessment of our work; we hope that our revisions to the Introduction further enhance its clarity.
>
> ### Weaknesses:
> *The proofs could be included in the supplementary material which would allow for more text interpreting the results and providing the intuition for their importance.*
>
> Thank you for this suggestion. We will defer all proofs to the Appendix, and expand the discussion around each result accordingly.
>
> *The authors should explain the solutions (15) and (16). The authors should explain in more detail their result, i.e. (18).*
>
> Thank you for these suggestions regarding the need to elaborate on our finite-ridge results. As mentioned in our common response, we will add more discussion on this point. Also, with regards to the interpretation of (15), we note that we point out in Lines 108-114 of the submitted manuscript that this solution is precisely the self-consistent equation for the limiting spectral moment generating function of the kernel matrix. This object is discussed in detail in previous works on the limiting spectra of product random matrices, e.g. ref. [28] in the submitted manuscript and references therein.
>
> *The authors should be more explicit about the notation, e.g. carefully check the use of /tilde or $\kappa_{0}$.*
>
> We will carefully proofread the manuscript to correct any notational issues.
>
> *In my opinion, the clarity of the paper could be improved.*
>
> Thank you for these suggestions regarding the clarity and explicitness of our manuscript. We hope the changes mentioned in the global response and elsewhere will help address your concerns; the extra page allowed for the camera-ready version will allow us to better unpack our results.
>
> ### Questions:
>
> *Do solutions (15) and (16) always exist? Can you elaborate on their form?*
>
> Solutions to these equations should exist for all covariance matrices with well-behaved limiting spectral densities, but they may not be writable in closed form. In the isotropic case, (16) is easily solvable; see the proof of Corollary 3.2. More generally, it must be solved numerically, as in previous works on ridge regression. Solving (15) is more challenging still; see the work of Penson and Zyczkowski referenced in our common response, and also ref. [28] in the submitted manuscript.
>
> *Do you have any intuition how quickly convergence to (13) happen?*
>
> To our knowledge, the best error estimates in the case (13) are known in the shallow setting from work by Cheng and Montanari (https://arxiv.org/abs/2210.08571), which shows convergence of generalization error with a multiplicative error that is roughly of order $1/n^{1-\epsilon}$ for some small positive $\epsilon$. We would conjecture, but have not attempted to prove, that a similar statement would hold in our setting.
>
> *Do you have any intuition about the limits when p is fixed?*
> Thank you for this question. If one considers a limit in which the widths of the hidden layers $n_{1}, \ldots, n_{L}$ are taken to infinity for a fixed number of training examples $p$, then the generalization error of the deep linear model should tend simply to that of the shallow model. This is shown by the large-width expansion in Corollary 3.5.

---

> > ### Comment · Reviewer_cGDX · 2023-08-17
> >
> > Thank you for the answers, I maintain my score.

---

### Official Review · Reviewer_hUXg · 2023-07-19

**Soundness:** 3 good
**Presentation:** 3 good
**Contribution:** 2 fair
**Rating:** 6
**Confidence:** 3

**Summary:**

This work studies the asymptotic risk of deep linear random features models (RFMs), with a high dimensional analysis based on the replica method. It extends previous work on this topic by relaxing the standard i.i.d assumption for the Gaussian weights in each layer.

Several consequences of their anaysis are discussed: (i)  several known results in linear regression, isotropic models, infinite width RFM, are recovered as special cases ; (ii) it is shown that feature anisotropy is detrimental, in the sense that the risk of the isotropic model  lowerbounds the risk of the general model ; (iii) it is shown that feature anisotropy does not affect the scaling laws of the risk ;  (iv) going beyond ridge regression, the analysis allows for the derivation of the risk of the Gibbs estimator, where feature anisotropy is shown to be generally beneficial (resp detrimental) for  large (resp small) prior variance.

**Strengths:**

* Sound piece of theoretical work.

* Relaxing the standard feature isotropy assumption seems to be a natural extension of the recent line of work on deep RFMs. Since the learned features of deep learning models often exhibit complex correlations, it could also provide insights on the effects of feature learning of deep networks in a controlled setting.

* The paper is very clearly written and enjoyable to read.

**Weaknesses:**

My main reservation regarding this paper is related to its scope -- and the significance of the results.

* While the analysis presented in the paper is novel and fills a gap in the literature by relaxing a standard assumption, the technical innovations appear to be somewhat limited and incremental compared to previous work, such as References [28].

I believe some related references were missed, see e.g., Mel & Pennington (ICLR 2022) -- which also investigates the effect of feature anisotropy  in random feature regression. I understand the main technical difference is that they work with shallow models with a Gaussian feature matrix, whereas the current paper work with Gaussian products (one may argue that the first setting is sufficient to capture the effects of anisotropy in RFMs).

* The insights gained from this analysis also seem to have certain limitations. For instance, considering  that in setting (13), the risk is studied in expectation over rotation invariant Gaussian matrices $Z_\ell$ at each layer, the result of Section 4 on the optimality of the isotropic case appears rather unsurprising to me.

Moreover I believe that the studied setting may not allow for  significant insights into representation learning (not that the authors claim otherwise, but this is one of the motivations for the work i.m.o).  For example , the assumption of Gaussians requires layerwise independence, while in feature learning scenarios, one would expect learned features in different layers to be correlated with each other -- and with the underlying data structure. So from that poinf of view, I feel the assumptions underlying this work are still quite restrictive.

On a minor related note, the general-sounding statement found in Section 1, "these results are consistent with the intuition that representation learning at only the first layer of a deep linear model is sufficient to achieve optimal performance" is a bit puzzing, as it seems to contradict known results in the topic, e.g. those on the implicit sparsity bias in deep linear networks (which requires representation learning in multiple layers, see e g., Woodworth et al, 2020).

**References**

Mel & Pennington (ICLR 2022). Anisotropic Random Feature Regression in High Dimensions. https://openreview.net/forum?id=JfaWawZ8BmX.
Woodworth et al, 2020. Kernel and rich regimes in overparametrized models, https://arxiv.org/abs/2002.09277.

**Questions:**

* As the authors mention in the conclusion,  Gaussian equivalence theorems could potentially extend these results to more general activations (and losses).  Including this extension would considerably increase the paper's scope. Could the authors provide a bit more details about the extent of the technical gaps to be filled to achieve this ?

* Could the authors comment on the comparisons with the results of Gerace et al, which seem to address the case of general RFMs with any (fixed) feature matrix F satisfying the balance condition (1.7)?

**References**

Gerace et al, 2020.  Generalisation error in learning with random features and the hidden manifold model. https://arxiv.org/pdf/2002.09339.pdf

**Limitations:**

Adequately acknowledged.

---

> ### Author Rebuttal · Authors · 2023-08-05
>
> We thank the referee for their careful and favorable assessment of our work. We hope that they find that our revisions will strengthen the paper.
>
> ### Weaknesses:
>
> *My main reservation regarding this paper is related to its scope [...]*
>
> Thank you for this comment. As we acknowledge, our work is a direct application of the formalism discussed in Ref. 28, which reflects the fundamental relationship between the generalization error of kernel ridge regression and the spectral statistics of the kernel matrix. However, our solutions in the ridgeless limit and detailed study of large-width properties are even in the context of the spectral properties novel.
>
> *I believe some related references were missed, see e.g., Mel & Pennington (ICLR 2022) [...]*
>
> We regret that we missed this reference in the submitted manuscript, and will add it to the revised version of our manuscript as part of an extended discussion of prior work on RFMs. To the best of our understanding, Mel and Pennington allows for anisotropic input correlations while still assuming uncorrelated weights, as defined in the prose between equations (2) and (3) of their ICLR paper. Moreover, they assume isotropic regularization in the definition of the regularized kernel matrix between equations (3) and (4). Therefore, if one has even a single layer of random features with anisotropic row and column correlations, our setting is more general.
>
> *The insights gained from this analysis also seem to have certain limitations [...] the result of Section 4 on the optimality of the isotropic case appears rather unsurprising to me.*
>
> We agree that rotation-invariance makes the optimality of isotropic weight spectra intuitive; this rotation-invariance also makes it intuitive why the dependence of the generalization error on the spectra should decouple across layers. We propose to add the following comment to Section 4 under Lemma 4.1 to note that “The fact that we study generalization in expectation over matrix-Gaussian random features makes the optimality of isotropic spectra intuitive. Since we can represent each structured weight matrix as a product of fixed covariance matrices multiplying an unstructured, rotation invariant Gaussian matrix, it makes sense that there should be no preferred directions along which variance can be concentrated to reduce error.” We hope that this addition will make this limitation of our analysis more explicit.
>
> *Moreover I believe that the studied setting may not allow for significant insights into representation learning [...].*
>
> We agree that the setting of our work is restricted, which facilitates our analytical progress. Moreover, we agree wholeheartedly that it would be interesting to address the case of correlated weights across layers. As a first step, this could perhaps be addressed within a model in which the weights are jointly Gaussian across layers. However, even if the weights within layers are uncorrelated, this is not directly addressable within the formalism used here, which relies heavily on the assumption of layer-wise independence. This limitation is shared by other physics-style approaches to product random matrix problems. We would therefore suggest that detailed study of such a model could be an interesting topic for future work, with the results presented here as a first step towards understanding RFMs with maximally general weight correlations within and across layers.
>
> *On a minor related note, the general-sounding statement found in Section 1 [...] seems to contradict known results in the topic [...].*
>
> We acknowledge that this statement, as written, is not sufficiently precise. We will revise it to read, “these results are consistent with the intuition that representation learning at only the first layer of a deep linear model is sufficient to recover a single teacher weight vector.” We appreciate the referee’s feedback on whether this revision addresses their concern, and will also cite Woodworth et al. in our updated manuscript. We note also that Woodworth et al. focus on optimization with gradient flow.
>
> ### Questions:
> *[...] Gaussian equivalence theorems could potentially extend these results to more general activations (and losses). [...]*
>
> Existing Gaussian equivalence theorems for deep nonlinear random feature models from Schröder et al. and Bosch et al. (both from 2023) depend on certain concentration and approximate orthogonality assumptions for feature Gram matrices (see eq. 11 of Schröder et al, https://arxiv.org/pdf/2302.00401). When the weights are independent across features, these properties can be verified. For weights drawn from a general matrix Gaussian, this does not obviously follow. To prove Gaussian equivalence for nontrivial covariance matrices, one would need to determine the class of matrices and of nonlinearities for which the required orthogonality conditions can be established. We would be happy to add a comment on this point to the discussion.
>
> *Could the authors comment on the comparisons with the results of Gerace et al [...]?*
>
> Thank you for this question. The results of Gerace et al depend on the solution to the set of self-consistent equations (2.4), which in turn depend on the limiting Stieltjes transform of the feature matrix. Thus, to apply their results to our setting, one would need to verify the balance condition holds for Gaussian product matrices with sufficiently high probability, and then compute the Stieltjes transform. The computation of the Stieljtes transform is a bottleneck: Gerace et al. focus on settings in which it is known. In our case, the Stieljtes transform of the random feature matrix would be determined by the solution to a non-trivial self-consistent equation of the form (15). Therefore, our replica theory approach in principle offers an alternative route to obtain what should be the same result. We will make sure to cite Gerace et al. in our updated manuscript, and to comment on this point.

---

> > ### Comment · Reviewer_hUXg · 2023-08-21
> >
> > I thank the authors for their responses.  I've also read other reviews and their rebuttal. I am raising my score.

---

### Official Review · Reviewer_bdDi · 2023-07-24

**Soundness:** 3 good
**Presentation:** 2 fair
**Contribution:** 2 fair
**Rating:** 5
**Confidence:** 4

**Summary:**

The paper derives the learning curves (generalization error vs number of samples) for random linear deep networks with Gaussian weights with non-trivial correlations. The correlations at the first layer control the performance of those networks, while all other layers (possibly of different widths) are equivalent to a single layer of minimal width among them.

**Strengths:**

*	Quality: the main result that in the noiseless case, correlations at the first layer improve performance, while correlations in the other layers degrade it (Lemmas 4.1 and 4.2) is interesting (and is not fully captured by the summary “representation learning at only the first layer of a deep linear model is sufficient to achieve optimal performance”). The additional result for generalization error of scale-free correlations, which converge with previously known results for uncorrelated weights, is very nice.

**Weaknesses:**

*	Quality: it is not justified why the authors focus on the “ridgeless limit”; can’t the minimal generalization error be achieved at a finite value of lambda? Why not?
The model studied seems to collapse for large alpha0 (figure 1 a vs b), but the phenomena is not explained.
 *	Clarity: as a theory-heavy paper, the authors could have done a better job in keeping the notation clear. p was not properly defined; I assumed it was the number of samples. The spectral moments generating function from eq 15,17 is introduced only at eq 19. The functions (or scalars) phi and phi bar (evaluated at k0) are not introduced. The important “expectation with respect to the limiting spectral distribution” is not properly defined (“is defined in eq X of the SM” would have been fine as well).
 *	Significance: the results of sections 4 and 5 seem to suggest the depth of the network does not contribute anything, as only alpha_min enters the results. This is probably a limitation of the linear model, and thus the discussion on the behaviour of deep networks seems empty. Also, the focus on the noiseless case makes it hard to extrapolate what (if any) of the results hold under noise at an optimal finite lambda.

**Questions:**

* Can the minimal generalization error be achieved at a finite lambda value?
 * Is there any effect of depth when alpha_min is kept fixed?

**Limitations:**

The authors are clear about the limitations of their work, where only the behaviour for the ridgeless case is studied (and hence the behaviour under noise is not well understood), and only linear networks are studied (and hence the depth does not play an effect), with random initialization (and hence there is no training).

---

> ### Author Rebuttal · Authors · 2023-08-05
>
> ### Strengths:
> *Quality: the main result that in the noiseless case, correlations at the first layer improve performance, while correlations in the other layers degrade it (Lemmas 4.1 and 4.2) is interesting (and is not fully captured by the summary “representation learning at only the first layer of a deep linear model is sufficient to achieve optimal performance”). The additional result for generalization error of scale-free correlations, which converge with previously known results for uncorrelated weights, is very nice.*
>
> Thank you for your careful reading of our paper, and for your positive assessment of its results. We will revise the summary sentence in the Introduction to: “These results are consistent with the intuition that representation learning at only the first layer of a deep linear model is sufficient to recover a single teacher weight vector..” We hope the referee agrees that this more clearly describes our result.
>
> ### Weaknesses:
>
> - *Quality: it is not justified why the authors focus on the “ridgeless limit”; can’t the minimal generalization error be achieved at a finite value of lambda? Why not? The model studied seems to collapse for large alpha0 (figure 1 a vs b), but the phenomena is not explained.*
>
> Thank you for this question. Please see our detailed discussion of this point in the common response to referees. The ‘collapse’ observed by the reviewer in Figure 1b reflects the fact that the generalization error of the model with power-law features diverges as $\alpha_{\ell} \downarrow 0$, which results from the fact that these spectra have diverging mean.
>
> - *Clarity: as a theory-heavy paper, the authors could have done a better job in keeping the notation clear. p was not properly defined; I assumed it was the number of samples. The spectral moments generating function from eq 15,17 is introduced only at eq 19. The functions (or scalars) phi and phi bar (evaluated at k0) are not introduced. The important “expectation with respect to the limiting spectral distribution” is not properly defined (“is defined in eq X of the SM” would have been fine as well).*
>
> Thank you for this suggestion.
> 1. We note that $p$ was defined in Line 50 of the submitted manuscript (“...with $p$ i.i.d. training examples…”). We recognize that there are varying conventions for the number of training samples in the literature, with $p$ being the common convention in physics and $n$ the usual choice for statisticians. We will add a footnote to emphasize this convention.
> 2.  The spectral generating functions are introduced in Equation 14, under Assumption 3.1 (Lines 89-92).
> 3. The function $\psi$ is defined in (14) as part of Assumption 3.1. Can the reviewer point us to where we have failed to define $\phi$?
> We will replace Lines 98-99 of the submitted manuscript with the following clarification: “...where $\mathbb{E}\_{\tilde{\sigma}\_{\ell}}[h(\tilde{\sigma}\_{\ell})] = \lim\_{n\_{\ell} \to \infty} \frac{1}{n\_{\ell}} \sum\_{j=1}^{n\_{\ell}} h(\tilde{\sigma}\_{\ell,j})$ denotes expectation of a function $h$ with respect to the limiting spectral distribution of $\tilde{\mathbf{\Sigma}}_{\ell}$, for $\{\tilde{\sigma}\_{\ell,j}\}$ its eigenvalues.”
> We hope these clarifications address your concerns. We will carefully go over the paper to address further points of clarification.
>
> - *Significance: the results of sections 4 and 5 seem to suggest the depth of the network does not contribute anything, as only alpha_min enters the results. This is probably a limitation of the linear model, and thus the discussion on the behaviour of deep networks seems empty. Also, the focus on the noiseless case makes it hard to extrapolate what (if any) of the results hold under noise at an optimal finite lambda.*
>
> Thank you for this comment. It is not correct to say that only $\alpha_{\mathrm{min}}$ enters the generalization error. This can be seen from the formula for the generalization error given in Proposition 3.2, equation (23): in the overparameterized regime, the relative widths $\alpha_{\ell}$ for the other layers enters into the functions $\mu_{\ell}$ defined in equation (21). Most simply, consider the isotropic case given in Corollary 3.2, equation (25): in the overparameterized regime $\alpha_{0}, \alpha_{\mathrm{min}} > 1$, the generalization error has explicit, obvious dependence on $\alpha_{\ell}$. This is also true of the subsequent corollaries in Sections 4 and 5. It is true that only the narrowest layer determines whether the model is overparameterized or bottlenecked, but the other layers have a definitely non-trivial effect. We elaborate on this issue under your subsequent Question regarding this issue. In regards to finite regularization, please see our common response.
>
> ### Questions:
> - *Can the minimal generalization error be achieved at a finite lambda value?*
>
> Please see our common response for a discussion of finite-$\lambda$ properties. In short, for the deep models we consider here it is challenging to analytically determine the optimal ridge..
>
> - *Is there any effect of depth when alpha_min is kept fixed?*
>
> Thank you for this question. As mentioned above, there is most definitely an effect of depth even if $\alpha_{\mathrm{min}}$ is held fixed. To potentially see this more clearly, consider the case in which all layers have the same width $\alpha_{1} = \alpha_{2} = \cdots = \alpha_{L}$. Then, in the overparameterized regime $\alpha_{0}, \alpha_{1} > 1$, we have $$\epsilon = \left(1 + \frac{L}{\alpha_{1} - 1}\right) \left(1 - \frac{1}{\alpha_{0}} \right) + \left(\frac{1}{\alpha_{0} - 1} + \frac{L}{\alpha_{1} -1}\right) \eta^2$$ Therefore, if one fixes $\alpha_{\mathrm{min}} = \alpha_{1}$ and increases the depth $L$, the generalization error will in this case increase linearly. Conceptually, this is due to the fact that increasing depth increases the variance of the random feature kernel. We will elaborate on this point in our updated manuscript.

---

> > ### Comment · Reviewer_bdDi · 2023-08-20
> > **Response to Rebuttal**
> >
> > Thanks for your clarifications and improved presentation. I would raise my score accordingly.

---

### Author Rebuttal · Authors · 2023-08-05

We thank the reviewers for their careful assessment of our work. A commonly-expressed concern was that the Abstract and Introduction were too terse to provide adequate guidance and context to the reader, and that the paper would be clearer if we provided more discussion of each result. For our revised manuscript, we will take advantage of the additional space freed up by deferring some of the proofs to the Appendix and the additional content page allowed for the camera-ready version to address these concerns.

In brief, we will make the following changes:
- We will expand the Abstract to explicitly mention the replica formalism and its applicability to the Bayesian setting, as well as our focus on interpolators and the power-law spectrum model. (Reviewer nhrG)
- We will add a more detailed description of previous work on random feature models to the Introduction, including the additional references suggested by Reviewers hUXg, nhrG, and VEZW. In particular, we will discuss the relevance of minimum-norm interpolation (i.e., the ridgeless limit $\lambda \downarrow 0$) with RFMs for questions of generalization in overparameterized models more generally, including deep networks. We will also expand our discussion of Gaussian equivalence results to better motivate our setup.
- We will add a discussion of the replica method and its applications to computing the generalization error of regression models to the Introduction. (Reviewer nhrG)
- We will convert our summary of contributions into a more readable bulleted list.
- In Section 3, when we introduce the general form for the generalization error that results from the replica computation (Proposition 3.1), we will expand our discussion of why we focus on the ridgeless limit. See below for a detailed discussion of this point.


We would like to elaborate on a common concern: our focus on ridgeless interpolators with $\lambda \downarrow 0$. Following the suggestions of Reviewer nhrG, we will discuss the relevance of minimum-norm interpolation with RFMs in the Introduction. In brief, recent interest in deep learning has focused on when interpolating regressors display benign overfitting properties, i.e., when they generalize well despite interpolating the training data. Moreover, we will add a discussion of the finite-$\lambda$ setting to Section 3 and to the Discussion. In brief, we agree with the referees (particularly Reviewer bdDi) that it is likely that the optimal generalization could be achieved at some non-zero value of $\lambda$. It is of course well-known from previous works that this holds for shallow ridge regression or a RFM with a single hidden layer is often finite and can even be negative, see for instance Wu and Xu, NeurIPS 2020 or Kobak et al., JMLR 2020. We will add a detailed discussion of this point to the updated manuscript.

However, how to obtain a clean form for the optimal ridge analytically is in the general-depth case less clear. If one considers differentiating the expression for the generalization error in Proposition 3.1 with respect to the explicit ridge, one would have terms corresponding to the derivatives of each of the inverse moment generating functions with respect to the self-consistently determined variable $\zeta$, and then the implicitly determined derivative of $\zeta$ itself from equation (15). It is not immediately clear to us whether a useful simplification immediately presents itself; we therefore are inclined to leave this question to more detailed investigation in future work. We remark also that the issue of solving (15) is related to a long line of previous work on finding the spectra of product random matrices. Here, even in the case in which all factors are isotropic, exact solutions are known only in the special case in which all factors are square (see Penson and Zyczkowski, Phys. Rev. E 2011). Even numerically, few detailed results are known for depths $L$ greater than 2 or 3. We therefore suggest to leave a detailed numerical investigation of optimal ridge to future work.

---

### Decision · Program_Chairs · 2023-09-21

**Decision:**

Accept (poster)

**Comment:**

The paper deals with the generalization of deep random feature models in the ridge regression setting. While the contribution is novel and interesting, the reviewers raised some limitations of the work, regarding its readability and its potential impact on the community in light of previous works. While the paper can be improved to be more impactful, its contribution is appreciated.